# A Simple and Adaptive Learning Rate for FTRL in Online Learning with Minimax Regret of $\Theta(T^{2/3})$ and its Application to Best-of-Both-Worlds

**Taira Tsuchiya**
The University of Tokyo and RIKEN
tsuchiya@mist.i.u-tokyo.ac.jp

**Shinji Ito**
The University of Tokyo and RIKEN
shinji@mist.i.u-tokyo.ac.jp

## Abstract

Follow-the-Regularized-Leader (FTRL) is a powerful framework for various online learning problems. By designing its regularizer and learning rate to be adaptive to past observations, FTRL is known to work adaptively to various properties of an underlying environment. However, most existing adaptive learning rates are for online learning problems with a minimax regret of $\Theta(\sqrt{T})$ for the number of rounds $T$, and there are only a few studies on adaptive learning rates for problems with a minimax regret of $\Theta(T^{2/3})$, which include several important problems dealing with indirect feedback. To address this limitation, we establish a new adaptive learning rate framework for problems with a minimax regret of $\Theta(T^{2/3})$. Our learning rate is designed by matching the stability, penalty, and bias terms that naturally appear in regret upper bounds for problems with a minimax regret of $\Theta(T^{2/3})$. As applications of this framework, we consider three major problems with a minimax regret of $\Theta(T^{2/3})$: partial monitoring, graph bandits, and multi-armed bandits with paid observations. We show that FTRL with our learning rate and the Tsallis entropy regularizer improves existing Best-of-Both-Worlds (BOBW) regret upper bounds, which achieve simultaneous optimality in the stochastic and adversarial regimes. The resulting learning rate is surprisingly simple compared to the existing learning rates for BOBW algorithms for problems with a minimax regret of $\Theta(T^{2/3})$.

## 1 Introduction

Online learning is a problem setting in which a learner interacts with an environment for $T$ rounds with the goal of minimizing their cumulative loss. This framework includes many important online decision-making problems, such as expert problems [21, 38, 57], multi-armed bandits [6, 8, 33], linear bandits [1, 14], graph bandits [4, 42], and partial monitoring [9, 11].

For the sake of discussion in a general form, we consider the following *general online learning framework*. In this framework, a learner is initially given a finite action set $\mathcal{A} = [k] := \{1, \dots, k\}$ and an observation set $\mathcal{O}$. At each round $t \in [T]$, the environment determines a loss function $\ell_t \colon \mathcal{A} \to [0, 1]$, and the learner selects an action $A_t \in \mathcal{A}$ based on past observations without knowing $\ell_t$. The learner then suffers a loss $\ell_t(A_t)$ and observes a feedback $o_t \in \mathcal{O}$. The goal of the learner is to minimize the (pseudo-)regret $\mathsf{Reg}_T$, which is defined as the expectation of the difference between the cumulative loss of the selected actions $(A_t)_{t=1}^T$ and that of an optimal action $a^* \in \mathcal{A}$ fixed in hindsight. That is, $\mathsf{Reg}_T = \mathbb{E}\left[\sum_{t=1}^T \ell_t(A_t) - \sum_{t=1}^T \ell_t(a^*)\right]$ for $a^* \in \arg\min_{a \in \mathcal{A}} \mathbb{E}\left[\sum_{t=1}^T \ell_t(a)\right]$. For example in the multi-armed bandit problem, the observation is $o_t = \ell_t(A_t)$.

38th Conference on Neural Information Processing Systems (NeurIPS 2024).

*Follow-the-Regularized-Leader (FTRL)* is a highly powerful framework for such online learning problems. In FTRL, a probability vector $q_t$ over $\mathcal{A}$, which is used for determining action selection probability $p_t$ so that $A_t \sim p_t$, is obtained by solving the following convex optimization problem:

$$q_t \in \operatorname*{arg\,min}_{q \in \mathcal{P}_k} \left\{ \sum_{s=1}^{t-1} \widehat{\ell}_s(q) + \beta_t \psi(q) \right\}, \tag{1}$$

where $\mathcal{P}_k$ is the set of probability distributions over $\mathcal{A} = [k]$, $\widehat{\ell}_t \colon \mathcal{P}_k \to \mathbb{R}$ is an estimator of loss function $\ell_t$, $\beta_t > 0$ is (a reciprocal of) learning rate at round $t$, and $\psi$ is a convex regularizer. FTRL is known for its usefulness in various online learning problems [1, 4, 8, 27, 37]. Notably, FTRL can be viewed as a generalization of Online Gradient Descent [63] and the Hedge algorithm [21, 38, 57], and is closely related to Online Mirror Descent [36, 45].

The benefit of FTRL due to its generality is that one can design its regularizer $\psi$ and learning rate $(\beta_t)_t$ so that it can perform adaptively to various properties of underlying loss functions. The *adaptive learning rate*, which exploits past observations, is often used to obtain such adaptivity. In order to see how it is designed, we consider the following stability–penalty decomposition, well-known in the literature [36, 45]:

$$\mathsf{Reg}_T \lesssim \underbrace{\sum_{t=1}^{T} \frac{z_t}{\beta_t}}_{\text{stability term}} + \underbrace{\beta_1 h_1 + \sum_{t=2}^{T} (\beta_t - \beta_{t-1}) h_t}_{\text{penalty term}}. \tag{2}$$

Intuitively, the *stability* term arises from the regret when the difference in FTRL outputs, $x_t$ and $x_{t+1}$, is large, and the *penalty* term is due to the strength of the regularizer. For example, in the Exp3 algorithm for multi-armed bandits [8], $h_t$ is the Shannon entropy of $x_t$ or its upper bound, and $z_t$ is the expectation of $(\nabla^2 \psi(x_t))^{-1}$-norm of the importance-weighted estimator $\widehat{\ell}_t$ or its upper bound.

Adaptive learning rates have been designed so that it depends on the stability or penalty. For example, the well-known AdaGrad [19, 44] and the first-order algorithm [2] depend on stability components $(z_s)_{s=1}^{t-1}$ to determine $\beta_t$. More recently, there are learning rates that depend on penalty components $(h_s)_{s=1}^{t-1}$ [25, 54] and that depend on both stability and penalty components [26, 28, 55].

However, almost all adaptive learning rates developed so far have been limited to problems with a minimax regret of $\Theta(\sqrt{T})$, and there has been limited investigation into problems with a minimax regret of $\Theta(T^{2/3})$ [25, 54]. Such online learning problems are primarily related to indirect feedback and includes many important problems, such as partial monitoring [9, 34], graph bandits [4], dueling bandits [51], online ranking [12], bandits with switching costs [18], and multi-armed bandits with paid observations [53]. The $\Theta(T^{2/3})$ problem is distinctive also due to the classification theorem in partial monitoring [9, 34, 35], which is a very general problem that includes a wide range of sequential decision-making problems as special cases. It is known that, the minimax regret of partial monitoring games can be classified into one of four categories: $0$, $\Theta(\sqrt{T})$, $\Theta(T^{2/3})$, or $\Omega(T)$. Among these, the classes with non-trivial difficulties and particular importance are the problems with a minimax regret of $\Theta(\sqrt{T})$ and $\Theta(T^{2/3})$.

**Contributions** To address this limitation, we establish a new learning rate framework for online learning with a minimax regret of $\Theta(T^{2/3})$. Henceforth, we will refer to problems with a minimax regret of $\Theta(T^{2/3})$ as *hard problems* to avoid repetition, abusing the terminology of partial monitoring. For hard problems, it is common to combine FTRL with *forced exploration* [4, 17, 34, 51]. In this study, we first observe that the regret of FTRL with forced exploration rate $\gamma_t$ is roughly bounded as follows:

$$\mathsf{Reg}_T \lesssim \underbrace{\sum_{t=1}^{T} \frac{z_t}{\beta_t \gamma_t}}_{\text{stability term}} + \underbrace{\beta_1 h_1 + \sum_{t=2}^{T} (\beta_t - \beta_{t-1}) h_t}_{\text{penalty term}} + \underbrace{\sum_{t=1}^{T} \gamma_t}_{\text{bias term}}. \tag{3}$$

Here, the third term, called the bias term, represents the regret incurred by forced exploration. In the aim of minimizing the RHS of (3), we will determine the exploration rate $\gamma_t$ and learning rate $\beta_t$ so that the above stability, penalty, and bias elements for each $t \in [T]$ are matched, where the resulting learning rate is called *Stability–Penalty–Bias matching learning rate (SPB-matching)*. This

**Table 1:** Regret bounds for partial monitoring, graph bandits, and multi-armed bandits (MAB) with paid observations. The number of rounds is denoted as $T$, the number of actions as $k$, and the minimum suboptimality gap as $\Delta_{\min}$. The variable $c_{\mathcal{G}}$ is defined in Section 5, $D$ is a constant dependent on the outcome distribution. The graph complexity measures $\delta, \delta^*$, satisfying $\delta^* \leq \delta$ for graphs with no self-loops, are defined in Section 6, and $\tilde{\delta}^* \leq \delta$ is the fractional weak domination number [13]. The parameter $c$ is the paid cost for observing a loss of actions. AwSB is the abbreviation of the adversarial regime with a self-bounding constraint. MS-type means that the bound in AdvSB has a form similar to the bound established by Masoudian and Seldin [43].

| Setting | Reference | Stochastic | Adversarial | AwSB |
|---|---|---|---|---|
| Partial monitoring (with global observability) | [30] | $D \log T$ | – | – |
| | [37] | – | $(c_{\mathcal{G}} T)^{2/3}(\log k)^{1/3}$ | – |
| | [54] | $\dfrac{c_{\mathcal{G}}^2 \log T \log(kT)}{\Delta_{\min}^2}$ | $(c_{\mathcal{G}} T)^{2/3}(\log T \log(kT))^{1/3}$ | ✓ |
| | [56] | $\dfrac{c_{\mathcal{G}}^2 k \log T}{\Delta_{\min}^2}$ | $(c_{\mathcal{G}} T)^{2/3}(\log T)^{1/3}$ | ✓ |
| | [15][a] | $\dfrac{c_{\mathcal{G}}^2 \log k \log T}{\Delta_{\min}^2}$ | $(c_{\mathcal{G}} T)^{2/3}(\log k)^{1/3}$ | ✓ |
| | **Ours (Cor. 9)** | $\dfrac{c_{\mathcal{G}}^2 \log k \log T}{\Delta_{\min}^2}$ | $(c_{\mathcal{G}} T)^{2/3}(\log k)^{1/3}$ | ✓ MS-type |
| Graph bandits (with weak observability) | [4] | – | $(\delta \log k)^{1/3} T^{2/3}$ | – |
| | [13] | – | $(\tilde{\delta}^* \log k)^{1/3} T^{2/3}$ | – |
| | [25] | $\dfrac{\delta \log T \log(kT)}{\Delta_{\min}^2}$ | $(\delta \log T \log(kT))^{1/3} T^{2/3}$ | ✓ |
| | [15][a,b] | $\dfrac{\delta \log k \log T}{\Delta_{\min}^2}$ | $(\delta \log k)^{1/3} T^{2/3}$ | ✓ |
| | **Ours (Cor. 11)** | $\dfrac{\delta^* \log k \log T}{\Delta_{\min}^2}$ | $(\delta^* \log k)^{1/3} T^{2/3}$ | ✓ MS-type |
| MAB with paid observations | [53] | – | $(ck \log k)^{1/3} T^{2/3} + \sqrt{T \log k}$ | – |
| | **Ours (Cor. 17)** | $\dfrac{\max\{c,1\} k \log k \log T}{\Delta_{\min}^2}$ | $(ck \log k)^{1/3} T^{2/3} + \sqrt{T \log k}$ | ✓ MS-type |

[a]The framework in [15] is a hierarchical reduction-based approach, rather than a direct FTRL method, discarding past observations as doubling-trick.

[b]The bounds in [15] depend on $\delta$, but their framework with the algorithm in [13] can achieve improved bounds replacing $\delta$ with $\tilde{\delta}^* \leq \delta$.

was inspired by the learning rate designed by matching the stability and penalty terms for problems with a minimax regret of $\Theta(\sqrt{T})$ [26]. Our learning rate is simultaneously adaptive to the stability component $z_t$ and penalty component $h_t$, which have attracted attention in very recent years [26, 28, 55]. The SPB-matching learning rate allows us to upper bound the RHS of (3) as follows:

**Theorem 1** (informal version of Theorem 6). *There exists learning rate $(\beta_t)_t$ and exploration rate $(\gamma_t)_t$ for which the RHS of (3) is bounded by $O\big(\big(\sum_{t=1}^{T} \sqrt{z_t h_t \log(\varepsilon T)}\big)^{2/3} + \big(\sqrt{z_{\max} h_{\max}}/\varepsilon\big)^{2/3}\big)$ for any $\varepsilon \geq 1/T$, where $z_{\max} = \max_{t \in [T]} z_t$ and $h_{\max} = \max_{t \in [T]} h_t$.*

Within the general online learning framework, this theorem allows us to prove the following Best-of-Both-Worlds (BOBW) guarantee [10, 58, 61], which achieves an $O(\log T)$ regret in the stochastic regime and an $O(T^{2/3})$ regret in the adversarial regime simultaneously:

**Theorem 2** (informal version of Theorem 7). *Under some regularity conditions, an FTRL-based algorithm with SPB-matching achieves $\mathsf{Reg}_T \lesssim (z_{\max} h_{\max})^{1/3} T^{2/3}$ in the adversarial regime. In the stochastic regime, if $\sqrt{z_t h_t} \leq \sqrt{\rho_1}(1 - q_{ta^*})$ holds for FTRL output $q_t \in \mathcal{P}_k$ and $\rho_1 > 0$ for all $t \in [T]$, the same algorithm achieves $\mathsf{Reg}_T \lesssim \frac{\rho_1}{\Delta_{\min}^2} \log(T \Delta_{\min}^3)$ for the minimum suboptimality gap $\Delta_{\min}$.*

To assess the usefulness of the above result that holds for the general online learning framework, this study focuses on two major hard problems: partial monitoring with global observability, graph

bandits with weak observability, and multi-armed bandits with paid observations. We demonstrate that the assumptions in Theorem 2 are indeed satisfied for these problems by appropriately choosing the parameters in SPB-matching, thereby improving the existing BOBW regret upper bounds in several respects. To obtain better bounds in this analysis, we leverage the smallness of stability components $z_t$, which results from the forced exploration. Additionally, SPB-matching is the first unified framework to achieve a BOBW guarantee for hard online learning problems. Our learning rate is based on a surprisingly simple principle, whereas existing learning rates for graph bandits and partial monitoring are extremely complicated (see [25, Eq. (15)] and [54, Eq. (16)]). Due to its simplicity, we believe that SPB-matching will serve as a foundation for building new BOBW algorithms for a variety of hard online learning problems.

The SPB-matching framework, though omitted from the main text due to the space constraints, is also applicable to the multi-armed bandits with paid observations [53], whose minimax regret with costs is $\Theta(T^{2/3})$. We can show that the regret with paid costs, $\mathsf{Reg}_T^{\mathsf{c}}$, is roughly bounded by $\mathsf{Reg}_T^{\mathsf{c}} = O\big((ck\log k)^{1/3}T^{2/3} + \sqrt{T\log k}\big)$ in the adversarial regime and $\mathsf{Reg}_T^{\mathsf{c}} = O\big(\max\{c,1\}k\log k\log T/\Delta_{\mathsf{min}}^2\big)$ in the stochastic regime for the cost of observation $c$. The bound for the adversarial regime is of the same order as [53, Theorem 3]. The detailed problem setup, regret upper bounds, and regret analysis can be found in Appendix G.

Although omitted in Theorem 2, our approach achieves a refined regret bound devised by Masoudian and Seldin [43] in the *adversarial regime with a self-bounding constraint* [61], which includes the stochastic regime, adversarial regime, and the stochastic regime with adversarial corruptions [41] as special cases. We call the refined bound *MS-type bound*, named after the author. The MS-type bound maintains an ideal form even when $C = \Theta(T)$ or $\Delta_{\mathsf{min}} = \Theta(1/\sqrt{T})$ (see [43] for details), and our bounds are the first MS-type bounds for hard problems. A comparison with existing regret bounds is summarized in Table 1.

## 2 Preliminaries

**Notation**  For a natural number $n \in \mathbb{N}$, we let $[n] = \{1, \ldots, n\}$. For vector $x$, let $x_i$ denote its $i$-th element and $\|x\|_p$ the $\ell_p$-norm for $p \in [1, \infty]$. Let $\mathcal{P}_k = \{p \in [0,1]^k : \|p\|_1 = 1\}$ be the $(k-1)$-dimensional probability simplex. The vector $e_i$ is the $i$-th standard basis and $\mathbf{1}$ is the all-ones vector. Let $D_\psi(x, y)$ denote the Bregman divergence from $y$ to $x$ induced by a differentiable convex function $\psi$: $D_\psi(x, y) = \psi(x) - \psi(y) - \langle \nabla \psi(y), x - y \rangle$. To simplify the notation, we sometimes write $(a_t)_{t=1}^T$ as $a_{1:T}$ and $f = O(g)$ as $f \lesssim g$. We regard function $f : \mathcal{A} = [k] \to \mathbb{R}$ as a $k$-dimensional vector.

**General online learning framework**  To provide results that hold for a wide range of settings, we consider the following general online learning framework introduced in Section 1.

> At each round $t \in [T] = \{1, \ldots, T\}$:
>   1. The environment determines a loss vector $\ell_t : \mathcal{A} \to [0, 1]$;
>   2. The learner selects an action $A_t \in \mathcal{A}$ based on $p_t \in \mathcal{P}_k$ without knowing $\ell_t$;
>   3. The learner suffers a loss of $\ell_t(A_t) \in [0, 1]$ and observes a feedback $o_t \in \mathcal{O}$.

This framework includes many problems such as the expert problem, multi-armed bandits, graph bandits, and partial monitoring as special cases.

**Stochastic, adversarial, and their intermediate regimes**  Within the above general online framework, we study three different regimes for a sequence of loss functions $(\ell_t)_t$. In the stochastic regime, the sequence of loss functions is sampled from an unknown distribution $\mathcal{D}$ in an i.i.d. manner. The suboptimality gap for action $a \in \mathcal{A}$ is given by $\Delta_a = \mathbb{E}_{\ell_t \sim \mathcal{D}}[\ell_t(a) - \ell_t(a^*)]$ and the minimum suboptimality gap by $\Delta_{\mathsf{min}} = \min_{a \neq a^*} \Delta_a$. In the adversarial regime, the loss functions can be selected arbitrarily, possibly based on the past history up to round $t-1$.

We also investigate, the adversarial regime with a self-bounding constraint [61], which is an intermediate regime between the stochastic and adversarial regimes.

**Definition 3.** Let $\Delta \in [0,1]^k$ and $C \geq 0$. The environment is in an *adversarial regime with a* $(\Delta, C, T)$ *self-bounding constraint* if it holds for any algorithm that $\mathsf{Reg}_T \geq \mathbb{E}\big[\sum_{t=1}^T \Delta_{A_t} - C\big]$.

From the definition, the stochastic and adversarial regimes are special cases of this regime. Additionally, the well-known stochastic regime with adversarial corruptions [41] also falls within this regime. For the adversarial regime with a self-bounding constraint, we assume that there exists a unique optimal action $a^*$. This assumption is standard in the literature of BOBW algorithms (*e.g.,* [22, 39, 58]).

## 3 SBP-matching: Simple and adaptive learning rate for hard problems

This section designs a new learning rate framework for hard online learning problems.

### 3.1 Objective function that adaptive learning rate aims to minimize

In hard problems, the regret of FTRL with somewhat large exploration rate $\gamma_t$ is known to be bounded in the following form [4, 25, 54]:

$$\text{Reg}_T \lesssim \sum_{t=1}^{T} \frac{z_t}{\beta_t \gamma_t} + \sum_{t=1}^{T} (\beta_t - \beta_{t-1}) h_t + \sum_{t=1}^{T} \gamma_t \tag{4}$$

for some stability component $z_t$ and penalty component $h_t$, where we set $\beta_{T+1} = \beta_T$ and $\beta_0 = 0$ for simplicity. Recall that the first term is the stability term, the second term is the penalty term, and the third term is the bias term, which arises from the forced exploration.

The goal when designing the adaptive learning rate is to minimize (4), under the constraints that $(\beta_t)_t$ is non-decreasing and $\beta_t$ depends on $(z_{1:t}, h_{1:t})$ or $(z_{1:t-1}, h_{1:t})$. A naive way to choose $\gamma_t$ to minimize (4) is to set $\gamma_t = \sqrt{z_t/\beta_t}$ so that the stability term and the bias term match. However, this choice does not work well in hard problems because to obtain a regret bound of (4), a lower bound of $\gamma_t \geq u_t/\beta_t$ for some $u_t > 0$ is needed. This lower bound is used to control the magnitude of the loss estimator $\widehat{\ell}_t$ (see *e.g.,* Eq. (61) for partial monitoring and Eq. (79) for graph bandits).[1] Therefore, we consider exploration rate of $\gamma_t = \gamma_t' + u_t/\beta_t$ for $\gamma_t' = \sqrt{z_t/\beta_t}$ and some $u_t > 0$, where $\gamma_t'$ is chosen so that the stability and bias terms are matched. With these choices,

$$\text{Eq. (4)} \leq \sum_{t=1}^{T} \left( \frac{z_t}{\beta_t \gamma_t'} + (\beta_t - \beta_{t-1}) h_t + \left( \gamma_t' + \frac{u_t}{\beta_t} \right) \right)$$

$$= \sum_{t=1}^{T} \left( 2\sqrt{\frac{z_t}{\beta_t}} + \frac{u_t}{\beta_t} + (\beta_t - \beta_{t-1}) h_t \right) =: F(\beta_{1:T}, z_{1:T}, u_{1:T}, h_{1:T}). \tag{5}$$

Note that the first two terms in $F$, $2\sqrt{z_t/\beta_t} + u_t/\beta_t$, come from the stability and bias terms and the last term, $(\beta_t - \beta_{t-1}) h_t$, is the penalty term. In the following, we investigate adaptive learning rate $(\beta_t)_{t=1}^{T}$ that minimizes $F$ in (5) instead of (4).

### 3.2 Stability–penalty–bias matching learning rate

We consider determining $(\beta_t)_t$ by matching the stability–bias terms and the penalty term as $2\sqrt{z_t/\beta_t} + u_t/\beta_t = (\beta_t - \beta_{t-1}) h_t$. Assume that when choosing $\beta_t$, we have an access to $\widehat{h}_t$ such that $h_t \leq \widehat{h}_t$.[2] Then, inspired by the above matching, we consider the following two update rules:

$$\text{(Rule 1)} \ \beta_t = \beta_{t-1} + \frac{1}{\widehat{h}_t} \left( 2\sqrt{\frac{z_t}{\beta_t}} + \frac{u_t}{\beta_t} \right), \quad \text{(Rule 2)} \ \beta_t = \beta_{t-1} + \frac{1}{\widehat{h}_t} \left( 2\sqrt{\frac{z_{t-1}}{\beta_{t-1}}} + \frac{u_{t-1}}{\beta_{t-1}} \right). \tag{6}$$

We call these update rules *Stability–Penalty–Bias Matching (SPB-matching)*. These are designed by following the simple principle of matching the stability, penalty, and bias elements, and Rules 1 and

---

[1]This is particularly the case when we use the Shannon entropy or Tsallis entropy regularizers, which is a weaker regularization than the log-barrier regularizer.

[2]In each problem setting, we will prove $h_t \leq c h_{t-1}$ for some constant $c$ (see Assumption (iii) in Theorem 7 and proofs of Theorems 8, 10 and 16). Hence if we set $\widehat{h}_t \leftarrow c h_{t-1}$, we have $h_t \leq \widehat{h}_t$. Note that $h_{t-1}$ can be calculated from the information available at the end of round $t-1$, and thus it can be used when determining $\beta_t$.

2 differ only in the way indices are shifted.[3] For the sake of convenience, we define $G_1$ and $G_2$ by

$$G_1(z_{1:T}, h_{1:T}) = \sum_{t=1}^{T} \frac{\sqrt{z_t}}{\left(\sum_{s=1}^{t} \sqrt{z_s}/h_s\right)^{1/3}}, \quad G_2(u_{1:T}, h_{1:T}) = \sum_{t=1}^{T} \frac{u_t}{\sqrt{\sum_{s=1}^{t} u_s/h_s}}. \quad (7)$$

Define $z_{\max} = \max_{t \in [T]} z_t$, $u_{\max} = \max_{t \in [T]} u_t$, and $h_{\max} = \max_{t \in [T]} h_t$. Then, using SPB-matching rules in (6), we can upper-bound $F$ in terms of $G_1$ and $G_2$ as follows:

**Lemma 4.** *Consider SPB-matching (6) and suppose that $h_t \leq \widehat{h}_t$ for all $t \in [T]$. Then, Rule 1 achieves $F(\beta_{1:T}, z_{1:T}, u_{1:T}, h_{1:T}) \leq 3.2 G_1(z_{1:T}, \widehat{h}_{1:T}) + 2 G_2(u_{1:T}, \widehat{h}_{1:T})$ and Rule 2 achieves $F(\beta_{1:T}, z_{1:T}, u_{1:T}, h_{1:T}) \leq 4 G_1(z_{1:T}, \widehat{h}_{2:T+1}) + 3 G_2(u_{1:T}, \widehat{h}_{2:T+1}) + 10\sqrt{z_{\max}/\beta_1} + 5 u_{\max}/\beta_1 + \beta_1 h_1$.*

The proof of Lemma 4 can be found in Appendix B.1. One can see from the proof that the effect of using $\gamma_t = \sqrt{z_t/\beta_t} + u_t/\beta_t$ instead of $\gamma_t = \sqrt{z_t/\beta_t}$ only appears in $G_2$, which has a less impact than $G_1$ when bounding $F$. We can further upper-bound $G_1$ as follows:

**Lemma 5.** *Let $(z_t)_{t=1}^{T} \subseteq \mathbb{R}_{\geq 0}$ and $(h_t)_{t=1}^{T} \subseteq \mathbb{R}_{>0}$ be any non-negative and positive sequences, respectively. Let $\theta_0 > \theta_1 > \cdots > \theta_J > \theta_{J+1} = 0$ and $\theta_0 \geq h_{\max}$ and define $\mathcal{T}_j = \{t \in [T] : \theta_{j-1} \geq h_t > \theta_j\}$ for $j \in [J]$ and $\mathcal{T}_{J+1} = \{t \in [T] : \theta_J \geq h_t\}$. Then, $G_1(z_{1:T}, h_{1:T}) \leq \frac{3}{2} \sum_{j=1}^{J+1} \left(\sqrt{\theta_{j-1}} \sum_{t \in \mathcal{T}_j} \sqrt{z_t}\right)^{2/3}$. This implies that for all $J \in \mathbb{N}$ it holds that*

$$G_1(z_{1:T}, h_{1:T}) \leq \frac{3}{2} \min\left\{ \left(\sqrt{2J} \sum_{t=1}^{T} \sqrt{z_t h_t}\right)^{\frac{2}{3}} + \left(2^{-J/2}\sqrt{z_{\max} h_{\max}}\right)^{\frac{2}{3}} T^{\frac{2}{3}}, \left(\sum_{t=1}^{T} \sqrt{z_t h_{\max}}\right)^{\frac{2}{3}} \right\}.$$

Combining Lemmas 4 and 5 and the bound on $G_2$ in [26, Lemma 3], we obtain the following theorem.

**Theorem 6.** *Let $(z_t)_{t=1}^{T}, (u_t)_{t=1}^{T} \subseteq \mathbb{R}_{\geq 0}$ and $(h_t)_{t=1}^{T} \subseteq \mathbb{R}_{>0}$. Suppose that $\widehat{h}_t$ satisfies $h_t \leq \widehat{h}_t$ for all $t \in [T]$. Then, if $\beta_t$ is given by Rule 1 in (6), then for all $\varepsilon \geq 1/T$ it holds that*

$$F(\beta_{1:T}, z_{1:T}, u_{1:T}, h_{1:T}) \lesssim \min\left\{ \left(\sum_{t=1}^{T} \sqrt{z_t \widehat{h}_t} \log(\varepsilon T)\right)^{\frac{2}{3}} + \left(\sqrt{z_{\max} \widehat{h}_{\max}}\big/\varepsilon\right)^{\frac{2}{3}}, \left(\sum_{t=1}^{T} \sqrt{z_t \widehat{h}_{\max}}\right)^{\frac{2}{3}} \right\}$$

$$+ \min\left\{ \sqrt{\sum_{t=1}^{T} u_t \widehat{h}_t \log(\varepsilon T)} + \sqrt{u_{\max} \widehat{h}_{\max}/\varepsilon}, \sqrt{\sum_{t=1}^{T} u_t \widehat{h}_{\max}} \right\}. \quad (8)$$

*If $\beta_t$ is given by Rule 2 in (6), then for all $\varepsilon \geq 1/T$ it holds that*

$$F(\beta_{1:T}, z_{1:T}, u_{1:T}, h_{1:T}) \lesssim \min\left\{ \left(\sum_{t=1}^{T} \sqrt{z_t \widehat{h}_{t+1}} \log(\varepsilon T)\right)^{\frac{2}{3}} + \left(\sqrt{z_{\max} \widehat{h}_{\max}}\big/\varepsilon\right)^{\frac{2}{3}}, \left(\sum_{t=1}^{T} \sqrt{z_t \widehat{h}_{\max}}\right)^{\frac{2}{3}} \right\}$$

$$+ \min\left\{ \sqrt{\sum_{t=1}^{T} u_t \widehat{h}_{t+1} \log(\varepsilon T)} + \sqrt{u_{\max} \widehat{h}_{\max}/\varepsilon}, \sqrt{\sum_{t=1}^{T} u_t \widehat{h}_{\max}} \right\} + \sqrt{\frac{z_{\max}}{\beta_1}} + \frac{u_{\max}}{\beta_1} + \beta_1 h_1. \quad (9)$$

Note that these bounds are for problems with a minimax regret of $\Theta(T^{2/3})$. Roughly speaking, our bounds have an order of $\left(\sum_{t=1}^{T} \sqrt{z_t \widehat{h}_{t+1}} \log T\right)^{1/3}$ and differ from the existing stability-penalty-adaptive-type bounds of $\sqrt{\sum_{t=1}^{T} z_t \widehat{h}_{t+1} \log T}$ for problems with a minimax regret of $\Theta(\sqrt{T})$ [26, 55]. We will see in the subsequent sections that our bounds are beneficial as they give nearly optimal regret bounds in stochastic and adversarial regimes in partial monitoring, graph bandits, and multi-armed bandits with paid observations.

**Algorithm 1:** Best-of-both-worlds framework based on FTRL with SPB-matching learning rate and Tsallis entropy for online learning with minimax regret of $\Theta(T^{2/3})$

---

1  **input:** action set $\mathcal{A}$, observation set $\mathcal{O}$, exponent of Tsallis entropy $\alpha$, $\beta_1$, $\bar{\beta}$
2  **for** $t = 1, 2, \ldots$ **do**
3  $\quad$ Compute $q_t \in \mathcal{P}_k$ by (10) with a loss estimator $\widehat{\ell}_t$.
4  $\quad$ Set $h_t = H_\alpha(q_t)$ and $z_t, u_t \geq 0$ defined for each problem.
5  $\quad$ Compute action selection probability $p_t$ from $q_t$ by (11).
6  $\quad$ Choose $A_t \in \mathcal{A}$ so that $\Pr[A_t = i \mid p_t] = p_{ti}$ and observe feedback $o_t \in \mathcal{O}$.
7  $\quad$ Compute loss estimator $\widehat{\ell}_t$ based on $p_t$ and $o_t$.
8  $\quad$ Compute $\beta_{t+1}$ by Rule 2 of SPB-matching in (6) with $\widehat{h}_{t+1} = h_t$.

---

## 4  Best-of-both-worlds framework for hard online learning problems

Using the SPB-matching learning rate established in Section 3, this section provides a BOBW algorithm framework for hard online learning problems. We consider the following FTRL update:

$$q_t = \underset{p \in \mathcal{P}_k}{\arg\min} \left\{ \sum_{s=1}^{t-1} \langle \widehat{\ell}_t, p \rangle + \beta_t(-H_\alpha(p)) + \bar{\beta}(-H_{\bar{\alpha}}(p)) \right\}, \quad \alpha \in (0, 1), \ \bar{\alpha} = 1 - \alpha, \quad (10)$$

where $H_\alpha$ is the $\alpha$-Tsallis entropy defined as $H_\alpha(p) = \frac{1}{\alpha} \sum_{i=1}^{k} (p_i^\alpha - p_i)$, which satisfies $H_\alpha(p) \geq 0$ and $H_\alpha(e_i) = 0$. Based on this FTRL output $q_t$, we set $h_t = H_\alpha(q_t)$, which satisfies $h_1 = h_{\max}$. Additionally, for $q_t$ and some $p_0 \in \mathcal{P}_k$, we use the action selection probability $p_t \in \mathcal{P}_k$ defined by

$$p_t = (1 - \gamma_t)q_t + \gamma_t\, p_0 \quad \text{for} \quad \gamma_t = \gamma_t' + \frac{u_t}{\beta_t} = \sqrt{\frac{z_t}{\beta_t}} + \frac{u_t}{\beta_t}, \quad (11)$$

where $\beta_1$ is chosen so that $\gamma_t \in [0, 1/2]$. Let $\kappa = \sqrt{z_{\max}/\beta_1} + u_{\max}/\beta_1 + \beta_1 h_1 + \bar{\beta}\bar{h}$ for $\bar{h} = H_{\bar{\alpha}}(\mathbf{1}/k)$ and let $\mathbb{E}_t[\cdot]$ be the expectation given all observations before round $t$. Then the above procedure with Rule 2 of SPB-matching in (6), summarized in Algorithm 1, achieves the following BOBW bound:

**Theorem 7.** *Consider the general online learning framework in Section 2 with $\|\ell_t\|_\infty \leq 1$. Suppose that Algorithm 1 satisfies the following three conditions (i)–(iii):*

$$\text{(i) } \mathsf{Reg}_T \leq \mathbb{E}\left[ \sum_{t=1}^{T} \langle \widehat{\ell}_t, q_t - e_{a^*} \rangle + 2 \sum_{t=1}^{T} \gamma_t \right],$$

$$\text{(ii) } \mathbb{E}_t\left[ \langle \widehat{\ell}_t, q_t - q_{t+1} \rangle - \beta_t D_{(-H_\alpha)}(q_{t+1}, q_t) \right] \lesssim \frac{z_t}{\beta_t \gamma_t'}, \quad \text{(iii) } h_t \lesssim h_{t-1}. \quad (12)$$

*Then, in the adversarial regime, Algorithm 1 achieves*

$$\mathsf{Reg}_T = O\left( (z_{\max} h_1)^{1/3} T^{2/3} + \sqrt{u_{\max} h_1 T} + \kappa \right). \quad (13)$$

*In the adversarial regime with a $(\Delta, C, T)$-self-bounding constraint, further suppose that*

$$\sqrt{z_t h_t} \leq \sqrt{\rho_1} \cdot (1 - q_{ta^*}) \quad \text{and} \quad u_t h_t \leq \rho_2 \cdot (1 - q_{ta^*}) \quad (14)$$

*are satisfied for some $\rho_1, \rho_2 > 0$ for all $t \in [T]$. Then, the same algorithm achieves*

$$\mathsf{Reg}_T = O\left( \frac{\rho}{\Delta_{\min}^2} \log\left(T\Delta_{\min}^3\right) + \left( \frac{C^2 \rho}{\Delta_{\min}^2} \log\left(\frac{T\Delta_{\min}}{C}\right) \right)^{1/3} + \kappa' \right) \quad (15)$$

*for $\rho = \max\{\rho_1, \rho_2\}$ and $\kappa' = \kappa + \left( (z_{\max} h_1)^{1/3} + \sqrt{u_{\max} h_1} \right)\left( 1/\Delta_{\min}^3 + C/\Delta_{\min} \right)^{2/3}$ when $T \geq 1/\Delta_{\min}^3 + C/\Delta_{\min} =: \tau$, and $\mathsf{Reg}_T = O\left( (z_{\max} h_1)^{1/3} \tau^{2/3} + \sqrt{u_{\max} h_1 \tau} \right)$ when $T < \tau$.*

The proof of Theorem 7 relies on Theorem 6 established in the last section and can be found in Appendix C. Note that the bound (15) becomes the bound for the stochastic regime when $C = 0$.

---

[3]The information available for determining $\beta_t$ differs between Rule 1 and Rule 2, and Rule 1 is included due to theoretical interest and will not be used after this section.

# 5 Case study (1): Partial monitoring with global observability

This section provides a new BOBW algorithm for globally observable partial monitoring games.

## 5.1 Problem setting and some concepts in partial monitoring

**Partial monitoring games** A Partial Monitoring (PM) game $\mathcal{G} = (\mathcal{L}, \Phi)$ consists of a loss matrix $\mathcal{L} \in [0,1]^{k \times d}$ and feedback matrix $\Phi \in \Sigma^{k \times d}$, where $k$ and $d$ are the number of actions and outcomes, respectively, and $\Sigma$ is the set of feedback symbols. The game unfolds over $T$ rounds between the learner and the environment. Before the game starts, the learner is given $\mathcal{L}$ and $\Phi$. At each round $t \in [T]$, the environment picks an outcome $x_t \in [d]$, and then the learner chooses an action $A_t \in [k]$ without knowing $x_t$. Then the learner incurs an unobserved loss $\mathcal{L}_{A_t x_t}$ and only observes a feedback symbol $\sigma_t := \Phi_{A_t x_t}$. This framework can be indeed expressed as the general online learning framework in Section 2, by setting $\mathcal{O} = \Sigma$, $\ell_t(a) = \mathcal{L}_{ax_t} = e_a^\top \mathcal{L} e_{x_t}$ and $o_t = \sigma_t = \Phi_{A_t x_t}$.

We next introduce fundamental concepts for PM games. Based on the loss matrix $\mathcal{L}$, we can decompose all distributions over outcomes. For each action $a \in [k]$, the cell of action $a$, denoted as $\mathcal{C}_a$, is the set of probability distributions over $[d]$ for which action $a$ is optimal. That is, $\mathcal{C}_a = \{u \in \mathcal{P}_d \colon \max_{b \in [k]} (\ell_a - \ell_b)^\top u \leq 0\}$, where $\ell_a \in \mathbb{R}^d$ is the $a$-th row of $\mathcal{L}$.

To avoid the heavy notions and concepts of PM, we assume that the PM game has no duplicate actions $a \neq b$ such that $\ell_a = \ell_b$ and its all actions are *Pareto optimal*; that is, $\dim(\mathcal{C}_a) = d - 1$ for all $a \in [k]$. The discussion of the effect of this assumption can be found *e.g.,* in [34, 37].

**Observability and loss estimation** Two Pareto optimal actions $a$ and $b$ are *neighbors* if $\dim(\mathcal{C}_a \cap \mathcal{C}_b) = d - 2$. Then, this neighborhood relations defines *globally observable games*, for which the minimax regret of $\Theta(T^{2/3})$ is known in the literature [9, 34]. Two neighbouring actions $a$ and $b$ are *globally observable* if there exists a function $w_{e(a,b)} \colon [k] \times \Sigma \to \mathbb{R}$ satisfying

$$\sum_{c=1}^{k} w_{e(a,b)}(c, \Phi_{cx}) = \mathcal{L}_{ax} - \mathcal{L}_{bx} \quad \text{for all } x \in [d], \tag{16}$$

where $e(a,b) = \{a, b\}$. A PM game is said to be globally observable if all neighboring actions are globally observable. To the end, we assume that $\mathcal{G}$ is globally observable.[4]

Based on the neighborhood relations, we can estimate the loss *difference* between actions, instead of estimating the loss itself. The *in-tree* is the edges of a directed tree with vertices $[k]$ and let $\mathscr{T} \subseteq [k] \times [k]$ be an in-tree over the set of actions induced by the neighborhood relations with an arbitrarily chosen root $r \in [k]$. Then, we can estimate the loss differences between Pareto optimal actions as follows. Let $G(a, \sigma)_b = \sum_{e \in \text{path}_{\mathscr{T}}(b)} w_e(a, \sigma)$ for $a \in [k]$, where $\text{path}_{\mathscr{T}}(b)$ is the set of edges from $b \in [k]$ to the root $r$ on $\mathscr{T}$. Then, it is known that this $G$ satisfies that for any Pareto optimal actions $a$ and $b$, $\sum_{c=1}^{k}(G(c, \Phi_{cx})_a - G(b, \Phi_{cx})_b) = \mathcal{L}_{ax} - \mathcal{L}_{bx}$ for all $x \in [d]$ (*e.g.,* [37, Lemma 4]). From this fact, one can see that we can use $\widehat{y}_t = G(A_t, \Phi_{A_t x_t})/p_{tA_t} \in \mathbb{R}^k$ as the loss (difference) estimator, following the standard construction of the importance-weighted estimator [8, 36]. In fact, $\widehat{y}_t$ satisfies $\mathbb{E}_{A_t \sim p_t}[\widehat{y}_{ta} - \widehat{y}_{tb}] = \sum_{c=1}^{k}(G(c, \sigma_t)_a - G(c, \sigma_t)_b) = \mathcal{L}_{ax} - \mathcal{L}_{bx}$. We let $c_{\mathcal{G}} = \max\{1, k\|G\|_\infty\}$ be a game-dependent constant, where $\|G\|_\infty = \max_{a \in [k], \sigma \in \Sigma} |G(a, \sigma)|$.

## 5.2 Algorithm and regret upper bounds

Here, we present a new BOBW algorithm based on Algorithm 1. We use the following parameters for Algorithm 1. We use the loss (difference) estimator of $\widehat{\ell}_t = \widehat{y}_t$. We set $p_0$ in (11) to $p_0 = \mathbf{1}/k$. For $\tilde{I}_t \in \arg\max_{i \in [k]} q_{ti}$ and $q_{t*} = \min\{q_{t\tilde{I}_t}, 1 - q_{t\tilde{I}_t}\}$, let

$$\beta_1 \geq \frac{64 c_{\mathcal{G}}^2}{1 - \alpha}, \quad \bar{\beta} = \frac{32 c_{\mathcal{G}} \sqrt{k}}{(1 - \alpha)^2 \sqrt{\beta_1}}, \quad z_t = \frac{4 c_{\mathcal{G}}^2}{1 - \alpha}\left(\sum_{i \neq \tilde{I}_t} q_{ti}^{2-\alpha} + q_{t*}^{2-\alpha}\right), \quad u_t = \frac{8 c_{\mathcal{G}}}{1 - \alpha} q_{t*}^{1-\alpha}. \tag{17}$$

Note that $z_{\max} = \frac{4 c_{\mathcal{G}}^2}{1-\alpha}$, $u_{\max} = \frac{8 c_{\mathcal{G}}}{1-\alpha}$, and $h_{\max} = h_1 = \frac{1}{\alpha} k^{1-\alpha}$. Then, we can prove the following:

**Theorem 8.** *In globally observable partial monitoring, for any $\alpha \in (0, 1)$, Algorithm 1 with (17) satisfies the assumptions of Theorem 7 with $\rho_1 = \Theta\left(\frac{c_{\mathcal{G}}^2 k^{1-\alpha}}{\alpha(1-\alpha)}\right)$ and $\rho_2 = \Theta\left(\frac{c_{\mathcal{G}} k^{1-\alpha}}{\alpha(1-\alpha)}\right)$.*

---

[4]Another representative class of PM is locally observable games, for which we can achieve a minimax regret of $\Theta(\sqrt{T})$. See [9, 36, 37] for local observability and [54, 55] for BOBW algorithms for it.

The proof of Theorem 8 is given in Appendix E. Setting $\alpha = 1 - 1/(\log k)$ gives the following:

**Corollary 9.** *In globally observable partial monitoring with $T \geq \tau$, Algorithm 1 with (17) for $\alpha = 1 - 1/(\log k)$ achieves*

$$
\mathsf{Reg}_T = \begin{cases} O\big((c_{\mathcal{G}}T)^{2/3}(\log k)^{1/3} + \kappa\big) & \text{in adversarial regime} \\[2ex] O\left(\dfrac{c_{\mathcal{G}}^2 \log k}{\Delta_{\min}^2} \log\big(T\Delta_{\min}^3\big) + \left(\dfrac{C^2 c_{\mathcal{G}}^2 \log k}{\Delta_{\min}^2} \log\left(\dfrac{T\Delta_{\min}}{C}\right)\right)^{1/3} + \kappa'\right) \\[2ex] \hspace{3cm} \text{in adversarial regime with a } (\Delta, C, T)\text{-self-bounding constraint}\,. \end{cases}
\tag{18}
$$

*Here, if we use $\beta_1 = 64c_{\mathcal{G}}^2/(1-\alpha)$, which satisfies (17), $\kappa = O(c_{\mathcal{G}}^2 \log k + k^{3/2}(\log k)^{5/2})$ and $\kappa' = \kappa + O((c_{\mathcal{G}}^{2/3}(\log k)^{1/3} + \sqrt{c_{\mathcal{G}} \log k})(\frac{1}{\Delta_{\min}^3} + \frac{C}{\Delta_{\min}})^{2/3})$.*

This regret upper bound is better than the bound based on FTRL in [54, 56] in both stochastic and adversarial regimes, notably by a factor of $\log T$ or $k$ in the stochastic regime. The bound for the adversarial regime with a $(\Delta, C, T)$-self-bounding constraint is the first MS-type bound in PM. The upper bounds for the adversarial regime and stochastic regime are optimal in terms of $T$ [9, 30]; however, even without considering BOBW guarantees, the optimality with respect to other variables $k, m$, and $d$ is unclear (cf. [36, Section 37.9]), and exploring this is an important direction for future work. As discussed in Section 1, employing the black-box reduction approach in [15] also allows us to achieve an upper bound of the same order as our upper bound. Nevertheless, as previously mentioned, the blackbox approach is a complicated approach involving multi-stage reductions and has the drawback of discarding past observations, similar to the doubling-trick. Hence, demonstrating that using the FTRL framework alone can achieve the same upper bound is a significant theoretical advancement.

## 6 Case study (2): Graph bandits with weak observability

This section presents a new BOBW algorithm for weakly observable graph bandits.

### 6.1 Problem setting and some concepts in graph bandits

**Problem setting** In the graph bandit problem, the learner is given a directed feedback graph $G = (V, E)$ with $V = [k]$ and $E \subseteq V \times V$. For each $i \in V$, let $N^{\mathsf{in}}(i) = \{j \in V : (j, i) \in E\}$ and $N^{\mathsf{out}}(i) = \{j \in V : (i, j) \in E\}$ be the in-neighborhood and out-neighborhood of vertex $i \in V$, respectively. The game proceeds as the general online learning framework provided in Section 2, with action set $\mathcal{A} = V$, loss function $\ell_t : V \to [0, 1]$, and observation $o_t = \{\ell_t(j) : j \in N^{\mathsf{out}}(I_t)\}$.

**Observability and domination number** Similar to partial monitoring, the minimax regret of graph bandits is characterized by the properties of the feedback graph $G$ [4]. A graph $G$ is *observable* if it contains no self-loops, $N^{\mathsf{in}}(i) \neq \emptyset$ for all $i \in V$. A graph $G$ is *strongly observable* if $i \in N^{\mathsf{in}}(i)$ or $V \setminus \{i\} \subseteq N^{\mathsf{in}}(i)$ for all $i \in V$. Then, a graph $G$ is *weakly observable* if it is observable but not strongly observable.[5] The minimax regret of the weakly observable is known to be $\Theta(T^{2/3})$.

The weak domination number characterizes precisely the minimax regret. The *weakly dominating set* $D \subseteq V$ is a set of vertices such that $\{i \in V : i \notin N^{\mathsf{out}}(i)\} \subseteq \bigcup_{i \in D} N^{\mathsf{out}}(i)$. Then, the *weak domination number* $\delta(G)$ of graph $G$ is the size of the smallest weakly dominating set. For weakly observable $G$, the minimax regret of $\tilde{\Theta}(\delta^{1/3}T^{2/3})$ is known [4]. Instead, our bound depends on the *fractional domination number* $\delta^*(G)$, defined by the optimal value of the following linear program:

$$
\text{minimize } \sum_{i \in V} x_i \quad \text{subject to} \quad \sum_{i \in N^{\mathsf{in}}(j)} x_i \geq 1 \; \forall j \in V \,, \; 0 \leq x_i \leq 1 \; \forall i \in V \,.
\tag{19}
$$

We use $(x_i^*)_{i \in V}$ to denote the optimal solution of (19) and define its normalized version $u \in \mathcal{P}_k$ by $u_i = x_i^* / \sum_{j \in V} x_j^*$. The advantage of using the fractional domination number mainly lies in its computational complexity; further details are provided in Appendix F.1.

---

[5]Similar to the locally observable games of partial monitoring, we can achieve an $O(\sqrt{T})$ regret for graph bandits with strong observability. See *e.g.,* [4] for details.

## 6.2 Algorithm and regret analysis

Here, we present a new BOBW algorithm based on Algorithm 1. We use the following parameters for Algorithm 1. We use the estimator $\widehat{\ell}_t \in \mathbb{R}^k$ defined by $\widehat{\ell}_{ti} = \frac{\ell_{ti}}{P_{ti}}\mathbb{1}[i \in N^{\mathsf{out}}(I_t)]$ for $P_{ti} = \sum_{j \in N^{\mathsf{in}}(i)} p_{tj}$, which is unbiased and has been employed in the literature [4, 13]. We set $p_0$ in (11) to $p_0 = u$. For $\tilde{I}_t \in \arg\max_{i \in [k]} q_{ti}$ and $q_{t*} = \min\{q_{t\tilde{I}_t}, 1 - q_{t\tilde{I}_t}\}$, let

$$\beta_1 \geq \frac{64\delta^*}{1-\alpha} \, , \, \bar{\beta} = \frac{32\sqrt{k\delta^*}}{(1-\alpha)^2\sqrt{\beta_1}} \, , \, z_t = \frac{4\delta^*}{1-\alpha}\left(\sum_{i \in V\setminus\{\tilde{I}_t\}} q_{ti}^{2-\alpha} + q_{t*}^{2-\alpha}\right) , \, u_t = \frac{8\delta^*}{1-\alpha}q_{t*}^{1-\alpha} . \quad (20)$$

Note that $z_{\max} = \frac{4\delta^*}{1-\alpha}$, $u_{\max} = \frac{8\delta^*}{1-\alpha}$, and $h_{\max} = h_1 = \frac{1}{\alpha}k^{1-\alpha}$. Then, we can prove the following:

**Theorem 10.** *In the weakly observable graph bandit problem, for any $\alpha \in (0,1)$, Algorithm 1 with (20) satisfies the assumptions of Theorem 7 with $\rho_1 = \rho_2 = \Theta\left(\frac{\delta^* k^{1-\alpha}}{\alpha(1-\alpha)}\right)$.*

The proof of Theorem 10 is given in Appendix F. Setting $\alpha = 1 - 1/(\log k)$ gives the following:

**Corollary 11.** *In weakly observable graph bandits with $T \geq \max\{\delta^*(\log k)^2, \tau\}$, Algorithm 1 with (20) for $\alpha = 1 - 1/(\log k)$ achieves*

$$\mathsf{Reg}_T = \begin{cases} O\left(\delta^{*1/3}T^{2/3}(\log k)^{1/3} + \kappa\right) & \textit{in adversarial regime} \\ O\left(\frac{\delta^* \log k}{\Delta_{\min}^2}\log(T\Delta_{\min}^3) + \left(\frac{C^2\delta^* \log k}{\Delta_{\min}^2}\log\left(\frac{T\Delta_{\min}}{C}\right)\right)^{1/3} + \kappa'\right) \\ & \textit{in adversarial regime with a } (\Delta, C, T)\textit{-self-bounding constraint}. \end{cases} \quad (21)$$

*Here, if we use $\beta_1 = 64\delta^*/(1-\alpha)$, which satisfies (20), $\kappa = O(\delta^* \log k + k^{3/2}(\log k)^{5/2})$ and $\kappa' = \kappa + O(((\delta^* \log k)^{1/3} + \sqrt{\delta^* \log k})(\frac{1}{\Delta_{\min}^3} + \frac{C}{\Delta_{\min}})^{2/3})$.*

Our bound is the first BOBW FTRL-based algorithm with the $O(\log T)$ bound in the stochastic regime, improving the existing best FTRL-based algorithm in [25]. Compared to the reduction-based approach in [15], the dependences on $T$ are the same. However, our bound unfortunately depends on the fractional domination number $\delta^*$ instead of the weak domination number $\delta$, which can be smaller than $\delta^*$. Roughly speaking, this comes from the use of Tsallis entropy instead of Shannon entropy employed for the existing BOBW bound [25]. The technical challenges of making our bound depend on $\delta$ instead of $\delta^*$ or the weak fractional domination number $\tilde{\delta}^*$ are further discussed in Appendix F.3. Still, we believe that our algorithm can perform better since the reduction-based algorithm discards past observations as the doubling trick. Furthermore, the bound for the adversarial regime with a $(\Delta, C, T)$-self-bounding constraint is the first MS-type bound in weakly observable graph bandits.

## 7 Conclusion and future work

In this work, we investigated hard online learning problems, that is online learning with a minimax regret of $\Theta(T^{2/3})$, and established a simple and adaptive learning rate framework called stability–penalty–bias matching (SPB-matching). The SPB-matching allows us to prove a regret bound of $\left(\sum_{t=1}^{T}\sqrt{z_t\widehat{h}_{t+1}\log T}\right)^{2/3}$ for the stability component $z_t$ and the penalty component $\widehat{h}_{t+1}$, which differs from the existing stability-penalty-adaptive-type bounds for problems with a minimax regret of $\Theta(\sqrt{T})$ [26, 55]. We showed that FTRL with the SPB-matching learning rate and the Tsallis entropy regularizer improves the existing BOBW regret bounds based on FTRL for two typical hard problems with indirect feedback, partial monitoring with global observability, graph bandits with weak observability. We also showed that the SPB-matching can be applied to derive the first BOBW regret bounds for multi-armed bandits with paid observations.

Interestingly, the optimal exponent of Tsallis entropy in these settings is $1 - 1/(\log k)$, suggesting the reasonableness of using Shannon entropy in existing algorithms for partial monitoring [37] and graph bandits [4]. Our learning rate is surprisingly simple compared to existing ones for hard problems [25, 54]. Hence, it is important future work to investigate whether this simplicity can be leveraged to apply SPB-matching to other hard problems, such as bandits with switching costs [18] and dueling bandits with Borda winner [51].

## Acknowledgments and Disclosure of Funding

The authors are grateful to the anonymous reviewers for their insightful feedback and constructive suggestions, which have helped to significantly improve the manuscript. TT was supported by JST ACT-X Grant Number JPMJAX210E and JSPS KAKENHI Grant Number JP24K23852.

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

# A Additional related work

**Best-of-both-worlds algorithms** The study of BOBW algorithms was initiated by Bubeck and Slivkins [10], who focused on multi-armed bandits. The motivation arises from the difficulty of determining in advance whether the underlying environment is stochastic or adversarial in real-world problems. Since then, BOBW algorithms have been extensively studied [7, 16, 22, 40, 46, 52], and recently, FTRL is the common approach for developing BOBW algorithms [24, 28, 60, 62]. One reason is by appropriately designing the learning rate and regularizer of FTRL, we can prove a BOBW guarantee for various problem settings. Another reason is that FTRL-based approaches not only perform well in both stochastic and adversarial regimes but also achieve favorable regret bounds in the adversarial regime with a self-bounding constraint, intermediate settings including stochastically constrained adversarial regime [58] and stochastic regime with adversarial corruptions [41]. This intermediate regime is particularly useful, considering that real-world problems often lie between purely stochastic and purely adversarial regimes.

This study is closely related to FTRL with the Tsallis entropy regularization. Tsallis entropy in online learning was introduced in [3, 5], and its significance for BOBW algorithms was established in [61]. In the multi-armed bandit problem, using the exponent of Tsallis entropy $\alpha = 1/2$ provides optimal upper bounds, up to logarithmic factors, in both stochastic and adversarial regimes [61]. However, in the graph bandits, where the dependence on $k$ is critical or in decoupled settings, optimal upper bounds can be achieved with $\alpha \neq 1/2$ [26, 32, 48, 59]. In this work, we demonstrate that using the exponent tofo $\alpha = 1 - 1/(\log k)$ for the number of actions $k$ results in favorable regret bounds, as shown in Corollaries 9 and 11.

**Partial monitoring** Partial monitoring [11, 47, 50] is a very general online decision-making framework and includes a wide range of problems such as multi-armed bandits, (utility-based) dueling bandits [23], online ranking [12], and dynamic pricing [29]. The characterization of the minimax regret in partial monitoring has been progressively understood through various studies. It is known that all partial monitoring games can be classified into trivial, easy, hard, and hopeless games, where their minimax regrets are $0$, $\Theta(\sqrt{T})$, $\Theta(T^{2/3})$ and $\Omega(T)$. For comprehensive literature, refer to [9] and the improved results presented in [34, 35]. The games for which we can achieve a regret bound of $O(T^{2/3})$ correspond to globally observable games.

There is limited research on BOBW algorithms for partial monitoring with global observability [54, 56]. The existing bounds exhibit suboptimal dependencies on $k$ and $T$, particularly in the stochastic regime, which comes from the use of the Shannon entropy or the log-barrier regularization. By employing Tsallis entropy, our algorithm is the first to achieve ideal dependencies on both $k$ and $T$. It remains uncertain whether our upper bound in the stochastic regime is optimal with respect to variables other than $T$. While there is an asymptotic lower bound for the stochastic regime [30], its coefficient is expressed as a complex optimization problem. Investigating this lower bound further is important future work.

**Graph bandits** The study on the graph bandit problem, which is also known as online learning with feedback graphs, was initiated by [42]. This problem includes several important problems such as the expert setting, multi-armed bandits, and label-efficient prediction. For example, considering a feedback graph with only self-loops, one can see that this corresponds to the multi-armed bandit problem. One of the most seminal studies on the graph bandit problem is by Alon et al. [4], who elucidated how the structure of the feedback graph influences its minimax regret. They demonstrated that the minimax regret is characterized by the observability of the feedback graph, introducing the notions of weakly observable graphs and strongly observable graphs. Of particular relevance to this study is the minimax regret of $\tilde{O}(\delta T^{2/3})$ for weakly observable graphs, where $\delta$ is the weak domination number and $\tilde{O}(\cdot)$ ignores logarithmic factors. Recently, this upper bound was improved to $\tilde{O}(\delta^* T^{2/3})$ by replacing the weak domination number with the fractional weak domination number $\tilde{\delta}^*$ [13].

There are several BOBW algorithms for graph bandits [15, 20, 25, 31, 49]. However, only a few of these studies consider the weakly observable setting [15, 25, 31]. The existing results based on FTRL rely on the domination number rather than the weak domination number [31] or exhibit poor dependence on $T$ [25, 31], and the best regret bound of them still exhibited a dependence on $T$ of

$(\log T)^2$ [25]. Our algorithm is the first FTRL-based algorithm in the weakly observable setting that achieves an $O(\log T)$ stochastic bound.

## B  Proofs for SPB-matching learning rate (Section 3)

### B.1  Proof of Lemma 4

*Proof of Lemma 4.* We first consider Rule 1 in (6). The learning rate $\beta_t$ is lower-bounded as

$$\beta_t^{3/2} \geq \beta_t^{1/2}\left(\beta_{t-1} + \frac{2}{\widehat{h}_t}\sqrt{\frac{z_t}{\beta_t}}\right) \geq \beta_{t-1}^{3/2} + \frac{2\sqrt{z_t}}{\widehat{h}_t} \geq 2\sum_{s=1}^{t}\frac{\sqrt{z_s}}{\widehat{h}_s}\,, \tag{22}$$

where the first inequality follows from the definition of $\beta_t$ in (6) and the second inequality from the fact that $(\beta_t)_t$ is non-decreasing. We also have

$$\beta_t^2 \geq \beta_t\left(\beta_{t-1} + \frac{1}{\widehat{h}_t}\frac{u_t}{\beta_t}\right) \geq \beta_{t-1}^{3/2} + \frac{u_t}{\widehat{h}_t} \geq \sum_{s=1}^{t}\frac{u_s}{\widehat{h}_s}\,. \tag{23}$$

Using the last two lower bounds on $\beta_t$, we can bound $F$ in (5) as

$$\begin{aligned}
F(\beta_{1:T}, z_{1:T}, u_{1:T}, h_{1:T}) &\leq \sum_{t=1}^{T}\left(2\sqrt{\frac{z_t}{\beta_t}} + \frac{u_t}{\beta_t} + (\beta_t - \beta_{t-1})\widehat{h}_t\right) \\
&\leq \sum_{t=1}^{T}\left(4\sqrt{\frac{z_t}{\beta_t}} + 2\frac{u_t}{\beta_t}\right) \\
&\leq 4\sum_{t=1}^{T}\sqrt{\frac{z_t}{\left(2\sum_{s=1}^{t}\sqrt{z_s}/\widehat{h}_s\right)^{1/3}}} + 2\sum_{t=1}^{T}\frac{u_t}{\sqrt{\sum_{s=1}^{t}u_t/\widehat{h}_t}} \\
&= 3.2G_1(z_{1:T}, \widehat{h}_{1:T}) + 2G_2(u_{1:T}, \widehat{h}_{1:T})\,, \tag{24}
\end{aligned}$$

where the second inequality follows from the definition of $\beta_t$ in (6) and the third inequality from (22) and (23). This completes the proof of the first statement in Lemma 4.

We next consider Rule 2 in (6). In this case, we can bound $F$ as follows:

$$\begin{aligned}
F(\beta_{1:T}, z_{1:T}, u_{1:T}, h_{1:T}) &\leq 2\sqrt{\frac{z_1}{\beta_1}} + \frac{u_1}{\beta_1} + \beta_1 h_1 + \sum_{t=2}^{T}\left(2\sqrt{\frac{z_t}{\beta_t}} + \frac{u_t}{\beta_t} + (\beta_t - \beta_{t-1})\widehat{h}_t\right) \\
&= 2\sqrt{\frac{z_1}{\beta_1}} + \frac{u_1}{\beta_1} + \beta_1 h_1 + \sum_{t=2}^{T}\left(2\sqrt{\frac{z_t}{\beta_t}} + \frac{u_t}{\beta_t} + 2\sqrt{\frac{z_{t-1}}{\beta_{t-1}}} + \frac{u_{t-1}}{\beta_{t-1}}\right) \\
&\leq \beta_1 h_1 + \sum_{t=1}^{T}\left(4\sqrt{\frac{z_t}{\beta_t}} + 2\frac{u_t}{\beta_t}\right)\,, \tag{25}
\end{aligned}$$

where the equality follows from (6).

We then first consider bounding $\sum_{t=1}^{T}\sqrt{z_t/\beta_t}$. We can lower-bound $\beta_t^{3/2}$ as

$$\beta_t^{3/2} \geq \beta_t^{1/2}\left(\beta_{t-1} + \frac{2}{\widehat{h}_t}\sqrt{\frac{z_{t-1}}{\beta_{t-1}}}\right) \geq \beta_{t-1}^{3/2} + \frac{2\sqrt{z_{t-1}}}{\widehat{h}_t} \geq \beta_1^{3/2} + 2\sum_{s=2}^{t}\frac{\sqrt{z_{s-1}}}{\widehat{h}_s} =: \left(\beta_t^{(1)}\right)^{3/2}, \tag{26}$$

where we define

$$\beta_t^{(1)} = \left(\beta_1^{3/2} + 2\sum_{s=2}^{t}\frac{\sqrt{z_{s-1}}}{\widehat{h}_s}\right)^{2/3} = \left(\beta_1^{3/2} + 2\sum_{s=1}^{t-1}\frac{\sqrt{z_s}}{\widehat{h}_{s+1}}\right)^{2/3} \leq \beta_t\,. \tag{27}$$

In the following, we will upper-bound $\sum_{t=1}^{T} \sqrt{z_t/\beta_t} \leq \sum_{t=1}^{T} \sqrt{z_t/\beta_t^{(1)}}$. Let $c = (1+\delta)^2$ for $\delta > 0$ and and we then define $\mathcal{S} = \{t \in [T]: \beta_{t+1}^{(1)} \leq c^2 \beta_t^{(1)}\}$ and $\mathcal{S}^{\mathsf{c}} = [T] \setminus \mathcal{S} = \{t \in [T]: \beta_{t+1}^{(1)} > c^2 \beta_t^{(1)}\}$. From these definitions, we have

$$\sum_{t \in \mathcal{S}^{\mathsf{c}}} \sqrt{\frac{z_t}{\beta_t^{(1)}}} \leq \sum_{t \in \mathcal{S}^{\mathsf{c}}} \sqrt{\frac{z_{\max}}{\beta_t^{(1)}}} \leq \sum_{s=0}^{\infty} \left(\frac{1}{c}\right)^s \sqrt{\frac{z_{\max}}{\beta_1}} \leq \frac{1}{1-1/c} \sqrt{\frac{z_{\max}}{\beta_1}}. \tag{28}$$

Hence, using the last inequality, we obtain

$$\begin{aligned}
\sum_{t=1}^{T} \sqrt{\frac{z_t}{\beta_t}} &\leq \sum_{t \in \mathcal{S}} \sqrt{\frac{z_t}{\beta_t^{(1)}}} + \sum_{t \in \mathcal{S}^{\mathsf{c}}} \sqrt{\frac{z_t}{\beta_t^{(1)}}} \\
&\leq c \sum_{t \in \mathcal{S}} \sqrt{\frac{z_t}{\beta_{t+1}^{(1)}}} + \frac{1}{1-1/c} \sqrt{\frac{z_{\max}}{\beta_1}} \\
&\leq c \sum_{t \in \mathcal{S}} \sqrt{\frac{z_t}{\left(2 \sum_{s=1}^{t} \sqrt{z_s}/\widehat{h}_{s+1}\right)^{2/3}}} + \frac{1}{1-1/c} \sqrt{\frac{z_{\max}}{\beta_1}} \\
&= \frac{c}{2^{1/3}} G_1(z_{1:T}, \widehat{h}_{2:T+1}) + \frac{c}{c-1} \sqrt{\frac{z_{\max}}{\beta_1}}, \tag{29}
\end{aligned}$$

where the third inequality follows from the definition of $\beta^{(1)}$ in (26).

We next bound $\sum_{t=1}^{T} u_t/\beta_t$. We can lower-bound $\beta_t^2$ as

$$\beta_t^2 \geq \beta_t \left(\beta_{t-1} + \frac{1}{\widehat{h}_t} \frac{u_{t-1}}{\beta_{t-1}}\right) \geq \beta_{t-1}^2 + \frac{u_{t-1}}{\widehat{h}_t} \geq \beta_1^2 + \sum_{s=2}^{t} \frac{u_{s-1}}{\widehat{h}_s} =: \left(\beta_t^{(2)}\right)^2, \tag{30}$$

where we define

$$\beta_t^{(2)} = \sqrt{\beta_1^2 + \sum_{s=2}^{t} \frac{u_{s-1}}{\widehat{h}_s}} = \sqrt{\beta_1^2 + \sum_{s=1}^{t-1} \frac{u_s}{\widehat{h}_{s+1}}} \leq \beta_t. \tag{31}$$

In the following, we will upper-bound $\sum_{t=1}^{T} u_t/\beta_t \leq \sum_{t=1}^{T} u_t/\beta_t^{(2)}$. Let us define $\mathcal{T} = \{t \in [T]: \beta_{t+1}^{(2)} \leq c\beta_t^{(2)}\}$ and $\mathcal{T}^{\mathsf{c}} = [T] \setminus \mathcal{T} = \{t \in [T]: \beta_{t+1}^{(2)} > c\beta_t^{(2)}\}$. From these definitions, we have

$$\sum_{t \in \mathcal{T}^{\mathsf{c}}} \frac{u_t}{\beta_t^{(2)}} \leq \sum_{t \in \mathcal{T}^{\mathsf{c}}} \frac{u_{\max}}{\beta_t^{(2)}} \leq \sum_{s=0}^{\infty} \left(\frac{1}{c}\right)^s \frac{u_{\max}}{\beta_1} \leq \frac{1}{1-1/c} \frac{u_{\max}}{\beta_1}. \tag{32}$$

Hence, using the last inequality, we obtain

$$\begin{aligned}
\sum_{t=1}^{T} \frac{u_t}{\beta_t} &\leq \sum_{t \in \mathcal{T}} \frac{u_t}{\beta_t^{(2)}} + \sum_{t \in \mathcal{T}^{\mathsf{c}}} \frac{u_t}{\beta_t^{(2)}} \\
&\leq c \sum_{t \in \mathcal{T}} \frac{u_t}{\beta_{t+1}^{(2)}} + \frac{1}{1-1/c} \frac{u_{\max}}{\beta_1} \\
&\leq c \sum_{t \in \mathcal{T}} \frac{u_t}{\sqrt{\sum_{s=1}^{t} u_s/\widehat{h}_{s+1}}} + \frac{1}{1-1/c} \frac{u_{\max}}{\beta_1} \\
&= c\, G_2(u_{1:T}, \widehat{h}_{2:T+1}) + \frac{c}{c-1} \frac{z_{\max}}{\beta_1}. \tag{33}
\end{aligned}$$

Finally, combining (25) with (29) and (33), we obtain

$$F(\beta_{1:T}, z_{1:T}, u_{1:T}, h_{1:T}) \leq 3.2c\, G_1(z_{1:T}, \widehat{h}_{2:T+1}) + 2c\, G_2(u_{1:T}, \widehat{h}_{2:T+1})$$

$$+ \frac{c}{c-1}\left(2\sqrt{\frac{z_{\max}}{\beta_1}} + \frac{u_{\max}}{\beta_1}\right) + \beta_1 h_1. \tag{34}$$

Setting $c = 1.25$ completes the proof. $\qquad\qquad\square$

## B.2 Proof of Lemma 5

Before proving Lemma 5, we prepare the following lemma, a variant of [45, Lemma 4.13].

**Lemma 12.** *Let $\mathcal{T} \subseteq [T] = \{1, \dots, T\}$ and $(x_t)_{t \in \mathcal{T}}$ be a non-negative sequence. Then,*

$$\sum_{t \in \mathcal{T}} \frac{x_t}{\left(\sum_{s \in [t] \cap \mathcal{T}} x_s\right)^{1/3}} \leq \frac{3}{2} \left(\sum_{t \in \mathcal{T}} x_t\right)^{2/3}. \tag{35}$$

*Proof.* Let $S_t = \sum_{s \in [t] \cup \mathcal{T}} x_s$. Then,

$$\frac{x_t}{\left(\sum_{s \in [t] \cap \mathcal{T}} x_s\right)^{1/3}} = \frac{x_t}{S_t^{1/3}} = \int_{S_{t-1}}^{S_t} S_t^{-1/3} \mathrm{d}z \leq \int_{S_{t-1}}^{S_t} z^{-1/3} \mathrm{d}z = \frac{3}{2}\left(S_t^{2/3} - S_{t-1}^{2/3}\right). \tag{36}$$

Summing up the last inequality over $\mathcal{T}$, we obtain

$$\sum_{t \in \mathcal{T}} \frac{x_t}{\left(\sum_{s \in [t] \cap \mathcal{T}} x_s\right)^{1/3}} = \frac{3}{2} \sum_{t \in \mathcal{T}} \left(S_t^{2/3} - S_{t-1}^{2/3}\right) \leq \frac{3}{2} S_T^{2/3}, \tag{37}$$

where the last inequality follows from the telescoping argument with the assumption that $x_t \geq 0$. $\square$

*Proof of Lemma 5.* We upper-bound $G_1$ as follows:

$$G_1(z_{1:T}, h_{1:T}) = \sum_{t=1}^{T} \frac{\sqrt{z_t}}{\left(\sum_{s=1}^{t} \sqrt{z_s}/h_s\right)^{1/3}} = \sum_{j=1}^{J+1} \sum_{t \in \mathcal{T}_j} \frac{\sqrt{z_t}}{\left(\sum_{s=1}^{t} \sqrt{z_s}/h_s\right)^{1/3}}$$

$$\leq \sum_{j=1}^{J+1} \sum_{t \in \mathcal{T}_j} \frac{\sqrt{z_t}}{\left(\sum_{s \in \mathcal{T}_j \cap [t]} \sqrt{z_s}/h_s\right)^{1/3}} \leq \sum_{j=1}^{J+1} \sum_{t \in \mathcal{T}_j} \frac{\sqrt{z_t}}{\left(\sum_{s \in \mathcal{T}_j \cap [t]} \sqrt{z_s}/\theta_{j-1}\right)^{1/3}}$$

$$= \sum_{j=1}^{J+1} \theta_{j-1}^{1/3} \sum_{t \in \mathcal{T}_j} \frac{\sqrt{z_t}}{\left(\sum_{s \in \mathcal{T}_j \cap [t]} \sqrt{z_s}\right)^{1/3}} \leq \frac{3}{2} \sum_{j=1}^{J+1} \left(\sqrt{\theta_{j-1}} \sum_{t \in \mathcal{T}_j} \sqrt{z_t}\right)^{2/3}, \tag{38}$$

where the last inequality follows from Lemma 12. This completes the proof of the first statement in Lemma 5. Setting $J = 0$ and $\theta_0 = h_{\max}$ in (38) yields that

$$G_1(z_{1:T}, h_{1:T}) \leq \frac{3}{2} \left(\sum_{t=1}^{T} \sqrt{z_t h_{\max}}\right)^{2/3}. \tag{39}$$

Setting $\theta_j = 2^{-j} h_{\max}$ for $j \in \{0\} \cup [J]$ in (38) also gives

$$G_1(z_{1:T}, h_{1:T}) \leq \frac{3}{2} \sum_{j=1}^{J+1} \left(\sqrt{\theta_{j-1}} \sum_{t \in \mathcal{T}_j} \sqrt{z_t}\right)^{2/3}$$

$$\leq \frac{3}{2} \sum_{j=1}^{J} \left(\sqrt{\frac{\theta_{j-1}}{\theta_j}} \sum_{t \in \mathcal{T}_j} \sqrt{z_t h_t}\right)^{2/3} + \frac{3}{2} \left(\sqrt{\theta_J} \sum_{t \in \mathcal{T}_J} \sqrt{z_t}\right)^{2/3}$$

$$= \frac{3}{2} \sum_{j=1}^{J} \left(\sqrt{2} \sum_{t \in \mathcal{T}_j} \sqrt{z_t h_t}\right)^{2/3} + \frac{3}{2} \left(2^{-J/2} \sum_{t \in \mathcal{T}_J} \sqrt{z_t h_{\max}}\right)^{2/3}$$

$$\leq \frac{3}{2} \left(\sqrt{2J} \sum_{j=1}^{J} \sum_{t \in \mathcal{T}_j} \sqrt{z_t h_t}\right)^{2/3} + \frac{3}{2} \left(2^{-J/2} \sum_{t \in \mathcal{T}_J} \sqrt{z_t h_{\max}}\right)^{2/3}$$

(Hölder's inequality)

$$\le \frac{3}{2}\left(\sqrt{2J}\sum_{t=1}^{T}\sqrt{z_t h_t}\right)^{2/3} + \frac{3}{2}\left(2^{-J/2}\sqrt{z_{\max}h_{\max}}\right)^{2/3}T^{2/3}, \qquad (40)$$

where the second inequality follows from $(x+y)^{2/3} \le x^{2/3} + y^{2/3}$ for $x, y \ge 0$. Combining the last inequality and (39) completes the proof of the second statement in Lemma 5. $\qquad\square$

## C   Proof for best-of-both-worlds analysis in general online learning framework (Theorem 7, Section 4)

This section provides the proof of Theorem 7.

*Proof.* From Assumption (i), the regret is bounded as

$$\mathsf{Reg}_T \le \mathbb{E}\left[\sum_{t=1}^{T}\langle \widehat{\ell}_t, q_t - e_{a^*}\rangle + 2\sum_{t=1}^{T}\gamma_t\right]. \qquad (41)$$

From the standard FTRL analysis in [36, Exercise 28.12], we obtain

$$\sum_{t=1}^{T}\langle \widehat{\ell}_t, q_t - e_{a^*}\rangle \le \sum_{t=1}^{T}\left(\left\langle \widehat{\ell}_t, q_t - q_{t+1}\right\rangle - \beta_t D_{(-H_\alpha)}(q_{t+1}, q_t) + (\beta_t - \beta_{t-1})h_t\right) + \bar{\beta}\bar{h}. \qquad (42)$$

Combining the last two inequalities, we obtain

$$\mathsf{Reg}_T \le \mathbb{E}\left[\sum_{t=1}^{T}\left(\left\langle \widehat{\ell}_t, q_t - q_{t+1}\right\rangle - \beta_t D_{(-H_\alpha)}(q_{t+1}, q_t) + (\beta_t - \beta_{t-1})h_t + 2\gamma_t\right) + \bar{\beta}\bar{h}\right]$$

$$\lesssim \mathbb{E}\left[\sum_{t=1}^{T}\left(\frac{z_t}{\beta_t \gamma_t'} + (\beta_t - \beta_{t-1})h_t + \gamma_t\right) + \bar{\beta}\bar{h}\right] \qquad \text{(Assumption (ii) in (12))}$$

$$\lesssim \mathbb{E}\left[\sum_{t=1}^{T}\left(\frac{z_t}{\beta_t \gamma_t'} + (\beta_t - \beta_{t-1})h_t + \gamma_t' + \frac{u_t}{\beta_t}\right) + \bar{\beta}\bar{h}\right] \qquad \text{(definition of $\gamma_t$ in (11))}$$

$$\lesssim \mathbb{E}\left[\sum_{t=1}^{T}\left(\sqrt{\frac{z_t}{\beta_t}} + \frac{u_t}{\beta_t} + (\beta_t - \beta_{t-1})h_{t-1}\right) + \bar{\beta}\bar{h}\right] \qquad \text{(definition of $\gamma_t'$ and Assumption (iii))}$$

$$\lesssim \mathbb{E}[F(\beta_{1:T}, z_{1:T}, u_{1:T}, h_{0:T-1})] + \bar{\beta}\bar{h}, \qquad (43)$$

where the last inequality follows from (5). Now, since $\beta_t$ follows Rule 2 in (6) with $\widehat{h}_t = h_{t-1}$, Eq. (9) in Theorem 6 gives

$$F(\beta_{1:T}, z_{1:T}, u_{1:T}, h_{0:T-1}) \lesssim \left(\sum_{t=1}^{T}\sqrt{z_t h_1}\right)^{\frac{2}{3}} + \sqrt{\sum_{t=1}^{T}u_t h_1} + \sqrt{\frac{z_{\max}}{\beta_1}} + \frac{u_{\max}}{\beta_1} + \beta_1 h_1, \qquad (44)$$

$$F(\beta_{1:T}, z_{1:T}, u_{1:T}, h_{0:T-1}) \lesssim \inf_{\varepsilon \ge 1/T}\left\{\left(\sum_{t=1}^{T}\sqrt{z_t h_t \log(\varepsilon T)}\right)^{\frac{2}{3}} + \left(\frac{\sqrt{z_{\max}h_1}}{\varepsilon}\right)^{\frac{2}{3}}\right.$$

$$\left. + \sqrt{\sum_{t=1}^{T}u_t h_t \log(\varepsilon T)} + \sqrt{\frac{u_{\max}h_1}{\varepsilon}}\right\} + \sqrt{\frac{z_{\max}}{\beta_1}} + \frac{u_{\max}}{\beta_1} + \beta_1 h_1. \qquad (45)$$

Hence, in the adversarial regime, combining (43) and (44) gives

$$\mathsf{Reg}_T \lesssim \mathbb{E}\left[\left(\sum_{t=1}^{T}\sqrt{z_t h_1}\right)^{2/3} + \sqrt{\sum_{t=1}^{T}u_t h_1}\right] + \kappa \le (z_{\max}h_1)^{1/3}T^{2/3} + \sqrt{u_{\max}h_1 T} + \kappa, \qquad (46)$$

where we recall that $\kappa = \sqrt{z_{\max}/\beta_1} + u_{\max}/\beta_1 + \beta_1 h_1 + \bar{\beta}\bar{h}$. This completes the proof of (13).

We next consider the adversarial regime with a $(\Delta, C, T)$-self-bounding constraint. For any $\varepsilon \geq 1/T$, combining (43) and (45) gives

$$
\mathrm{Reg}_T \lesssim \mathbb{E}\left[\left(\sum_{t=1}^{T} \sqrt{z_t h_t} \log(\varepsilon T)\right)^{\frac{2}{3}} + \sqrt{\sum_{t=1}^{T} u_t h_t \log(\varepsilon T)}\right] + \left(\frac{\sqrt{z_{\max} h_1}}{\varepsilon}\right)^{\frac{2}{3}} + \sqrt{\frac{u_{\max} h_1}{\varepsilon}} + \kappa
$$

$$
\leq \left(\mathbb{E}\left[\sum_{t=1}^{T} \sqrt{z_t h_t}\right]\sqrt{\log(\varepsilon T)}\right)^{\frac{2}{3}} + \sqrt{\mathbb{E}\left[\sum_{t=1}^{T} u_t h_t\right]\log(\varepsilon T)} + \left(\frac{\sqrt{z_{\max} h_1}}{\varepsilon}\right)^{\frac{2}{3}} + \sqrt{\frac{u_{\max} h_1}{\varepsilon}} + \kappa,
$$

(47)

where the last inequality follows from Jensen's inequality. Now, using the assumption (14) and defining $Q(a^*) = \mathbb{E}\left[\sum_{t=1}^{T}(1 - q_{ta^*})\right] \in [0, T]$, we have

$$
\mathbb{E}\left[\sum_{t=1}^{T} \sqrt{z_t h_t}\right] \leq \sqrt{\rho_1}\, \mathbb{E}\left[\sum_{t=1}^{T}(1 - q_{ta^*})\right] = \sqrt{\rho_1}\, Q(a^*), \tag{48}
$$

$$
\mathbb{E}\left[\sum_{t=1}^{T} u_t h_t\right] \leq \rho_2\, \mathbb{E}\left[\sum_{t=1}^{T}(1 - q_{ta^*})\right] = \rho_2\, Q(a^*). \tag{49}
$$

Since we consider the adversarial regime with a $(\Delta, C, T)$-self-bounding constraint, the regret is lower-bounded as

$$
\mathrm{Reg}_T \geq \mathbb{E}\left[\sum_{t=1}^{T} \langle \Delta, p \rangle\right] - C \geq \frac{1}{2}\mathbb{E}\left[\sum_{t=1}^{T} \langle \Delta, q \rangle\right] - C
$$

$$
\geq \frac{1}{2}\Delta_{\min}\mathbb{E}\left[\sum_{t=1}^{T}(1 - q_{ta^*})\right] - C = \frac{1}{2}\Delta_{\min}Q(a^*) - C, \tag{50}
$$

where the second inequality follows from $p = (1 - \gamma_t)q_t + \gamma_t p_0 \geq q_t/2$. Hence, combining (47) with (48), (49) and (50), we can bound the regret for any $\lambda \in (0, 1]$ as follows:

$$
\mathrm{Reg}_T = (1 + \lambda)\mathrm{Reg}_T - \lambda\mathrm{Reg}_T
$$

$$
\lesssim (1 + \lambda)\left(\sqrt{\rho_1}Q(a^*)\sqrt{\log(\varepsilon T)}\right)^{2/3} - \frac{\lambda}{4}\Delta_{\min}Q(a^*) + (1 + \lambda)\sqrt{\rho_2 Q(a^*)\log(\varepsilon T)} - \frac{\lambda}{4}\Delta_{\min}Q(a^*)
$$

$$
+ (1 + \lambda)\left(\left(\frac{\sqrt{z_{\max} h_1}}{\varepsilon}\right)^{2/3} + \sqrt{\frac{u_{\max} h_1}{\varepsilon}} + \kappa\right) + \lambda C
$$

$$
\lesssim \frac{(1 + \lambda)^3}{\lambda^2}\frac{\rho_1 \log(\varepsilon T)}{\Delta_{\min}^2} + \frac{(1 + \lambda)^2}{\lambda}\frac{\rho_2 \log(\varepsilon T)}{\Delta_{\min}} + \left(\frac{\sqrt{z_{\max} h_1}}{\varepsilon}\right)^{2/3} + \sqrt{\frac{u_{\max} h_1}{\varepsilon}} + \kappa + \lambda C
$$

$$
\lesssim \frac{\rho_1 \log(\varepsilon T)}{\Delta_{\min}^2} + \frac{\rho_2 \log(\varepsilon T)}{\Delta_{\min}} + \frac{1}{\lambda^2}\left(\frac{\rho_1 \log(\varepsilon T)}{\Delta_{\min}^2} + \frac{\rho_2 \log(\varepsilon T)}{\Delta_{\min}}\right) + \left(\frac{\sqrt{z_{\max} h_1}}{\varepsilon}\right)^{2/3} + \sqrt{\frac{u_{\max} h_1}{\varepsilon}} + \kappa + \lambda C
$$

$$
\lesssim \frac{\rho \log(\varepsilon T)}{\Delta_{\min}^2} + \frac{1}{\lambda^2}\frac{\rho \log(\varepsilon T)}{\Delta_{\min}^2} + \left(\frac{\sqrt{z_{\max} h_1}}{\varepsilon}\right)^{2/3} + \sqrt{\frac{u_{\max} h_1}{\varepsilon}} + \kappa + \lambda C, \tag{51}
$$

where in the first inequality we used (47) with (48), (49), (50), and Jensen's inequality, in the second inequality we used $ax^2 - bx^3 \leq 4a^3/(27b^2)$ for $a \geq 0, b > 0$ and $x \geq 0$ and $ax - bx^2 \leq a^2/(4b)$ for $a \geq 0, b > 0$ and $x \geq 0$ and in the third inequality we used $\lambda \in (0, 1]$. Setting $\lambda = \Theta\left((\rho \log(\varepsilon T)/C)^{1/3}\right)$ in the last inequality, we obtain

$$
\mathrm{Reg}_T \lesssim \frac{\rho \log(\varepsilon T)}{\Delta_{\min}^2} + \left(\frac{C^2 \rho \log(\varepsilon T)}{\Delta_{\min}^2}\right)^{1/3} + \left(\frac{\sqrt{z_{\max} h_1}}{\varepsilon}\right)^{2/3} + \sqrt{\frac{u_{\max} h_1}{\varepsilon}} + \kappa.
$$

Finally, when $T \geq \tau = 1/\Delta_{\min}^3 + C/\Delta_{\min}$, setting

$$
\varepsilon = \frac{1}{\rho^2/\Delta_{\min}^3 + C\rho/\Delta_{\min}} \geq \frac{1}{T} \tag{52}
$$

yields

$$\mathsf{Reg}_T \lesssim \frac{\rho}{\Delta_{\min}^2} \log_+\left(\frac{T}{1/\Delta_{\min}^3 + C/\Delta_{\min}}\right) + \left(\frac{C^2\rho}{\Delta_{\min}^2} \log_+\left(\frac{T}{1/\Delta_{\min}^3 + C/\Delta_{\min}}\right)\right)^{1/3}$$

$$+ (z_{\max} h_1)^{1/3}\left(\frac{1}{\Delta_{\min}^3} + \frac{C}{\Delta_{\min}}\right)^{2/3} + \sqrt{u_{\max} h_1}\sqrt{\frac{1}{\Delta_{\min}^3} + \frac{C}{\Delta_{\min}}} + \kappa$$

$$\lesssim \frac{\rho}{\Delta_{\min}^2} \log_+\left(T\Delta_{\min}^3\right) + \left(\frac{C^2\rho}{\Delta_{\min}^2} \log_+\left(\frac{T\Delta_{\min}}{C}\right)\right)^{1/3}$$

$$+ \left((z_{\max} h_1)^{1/3} + \sqrt{u_{\max} h_1}\right)\left(\frac{1}{\Delta_{\min}^3} + \frac{C}{\Delta_{\min}}\right)^{2/3} + \kappa\,, \tag{53}$$

which completes the proof. $\qquad\square$

# D   Auxiliary lemmas

This section provides auxiliary lemmas useful for proving the BOBW gurantee.

**Lemma 13.** *Let $\alpha \in (0,1)$ and $i^* \in [k]$. Then, the $\alpha$-Tsallis entropy $H_\alpha$ is bounded from above as*

$$H_\alpha(q) = \frac{1}{\alpha}\sum_{i=1}^{k}(q_i^\alpha - q_i) \leq \frac{1}{\alpha}(k-1)^\alpha(1 - q_{i^*})^\alpha \tag{54}$$

*for any $q \in \mathcal{P}_k$.*

*Proof.* From Jensen's inequality and the fact that $x \mapsto x^\alpha$ is concave for $\alpha \in (0,1)$,

$$\sum_{i=1}^{k}(q_i^\alpha - q_i) \leq \sum_{i\neq i^*} q_i^\alpha = (k-1)\sum_{i\neq i^*}\frac{1}{k-1}q_i^\alpha \leq (k-1)\left(\frac{1}{k-1}\sum_{i\neq i^*}q_i\right)^\alpha$$

$$= (k-1)^{1-\alpha}\left(\sum_{i\neq i^*}q_i\right)^\alpha = (k-1)^{1-\alpha}(1 - q_{i^*})^\alpha\,, \tag{55}$$

which completes the proof. $\qquad\square$

**Lemma 14** ([26, Lemma 10]). *Let $q \in \mathcal{P}_k$ and $\tilde{I} \in \arg\max_{i\in[k]} q_i$. For $\ell \in \mathbb{R}^k$, if $|\ell_i| \leq \frac{1-\alpha}{4}\frac{1}{\min\{q_{\tilde{I}}, 1-q_{\tilde{I}}\}^{1-\alpha}}$ for all $i \in [k]$, it holds that*

$$\max_{p\in\mathcal{P}_k}\left\{\langle\ell, q-p\rangle - D_{(-H_\alpha)}(p, q)\right\} \leq \frac{4}{1-\alpha}\left(\sum_{i\neq\tilde{I}} q_i^{2-\alpha}\ell_i^2 + \min\{q_{\tilde{I}}, 1-q_{\tilde{I}}\}^{2-\alpha}\ell_{\tilde{I}}^2\right). \tag{56}$$

**Lemma 15** ([26, Lemmas 11 and 12]). *Let $L \in \mathbb{R}^k$ and $\ell \in \mathbb{R}^k$ and suppose that $q, r \in \mathcal{P}_k$ are given by*

$$q \in \arg\min_{p\in\mathcal{P}_k}\left\{\langle L, p\rangle + \beta(-H_\alpha(p)) + \bar{\beta}(-H_{\bar{\alpha}}(p))\right\}$$

$$r \in \arg\min_{p\in\mathcal{P}_k}\left\{\langle L+\ell, p\rangle + \beta'(-H_\alpha(p)) + \bar{\beta}(-H_{\bar{\alpha}}(p))\right\} \tag{57}$$

*for the Tsallis entropy $H_\alpha$ and $H_{\bar{\alpha}}$, $0 < \beta \leq \beta'$. Suppose also that*

$$\|\ell\|_\infty \leq \max\left\{\frac{1-(\sqrt{2})^{\alpha-1}}{2}q_*^{\alpha-1}\beta, \frac{1-(\sqrt{2})^{\bar{\alpha}-1}}{2}q_*^{\bar{\alpha}-1}\bar{\beta}\right\}, \tag{58}$$

$$0 \leq \beta' - \beta \leq \max\left\{\left(1-(\sqrt{2})^{\alpha-1}\right)\beta, \frac{1-(\sqrt{2})^{\bar{\alpha}-1}}{\sqrt{2}}q_*^{\bar{\alpha}-\alpha}\bar{\beta}\right\}. \tag{59}$$

*Then, it holds that $H_\alpha(r) \leq 2H_\alpha(q)$.*

# E  Proof for partial monitoring (Theorem 8, Section 5)

This section provides the proof of Theorem 8.

*Proof of Theorem 8.* It suffices to prove that assumptions in Theorem 7 are satisfied. We first vertify Assumptions (i)–(iii) in (12). Let us start by checking Assumption (i). From the definition of the loss difference estimator $\widehat{y}_t$, the regret is bounded as

$$
\begin{aligned}
\mathrm{Reg}_T &= \mathbb{E}\left[\sum_{t=1}^{T}(\mathcal{L}_{A_t x_t} - \mathcal{L}_{a^* x_t})\right] = \mathbb{E}\left[\sum_{t=1}^{T}\langle p_t - e_{a^*}, \mathcal{L}e_{x_t}\rangle\right] \\
&= \mathbb{E}\left[\sum_{t=1}^{T}\langle q_t - e_{a^*}, \mathcal{L}e_{x_t}\rangle + \sum_{t=1}^{T}\gamma_t\left\langle \frac{1}{k}\mathbf{1} - q_t, \mathcal{L}e_{x_t}\right\rangle\right] \\
&\leq \mathbb{E}\left[\sum_{t=1}^{T}\langle q_t - e_{a^*}, \mathcal{L}e_{x_t}\rangle + \sum_{t=1}^{T}\gamma_t\right] = \mathbb{E}\left[\sum_{t=1}^{T}\sum_{a=1}^{k}q_{ta}(\mathcal{L}_{ax_t} - \mathcal{L}_{a^* x_t}) + \sum_{t=1}^{T}\gamma_t\right] \\
&= \mathbb{E}\left[\sum_{t=1}^{T}\sum_{a=1}^{k}q_{ta}(\widehat{y}_{ta} - \widehat{y}_{ta^*}) + \sum_{t=1}^{T}\gamma_t\right] = \mathbb{E}\left[\sum_{t=1}^{T}\langle q_t - e_{a^*}, \widehat{y}_t\rangle + \sum_{t=1}^{T}\gamma_t\right],
\end{aligned} \tag{60}
$$

where the inequality holds since $\mathcal{L} \in [0, 1]^{k \times d}$, This implies that Assumption (i) is indeed satisfied.

We next check Assumption (ii) in (12). For any $b \in [k]$ we have

$$
\left|\frac{\widehat{y}_{tb}}{\beta_t}\right| = \left|\frac{G(A_t, \sigma_t)_b}{\beta_t p_{tA_t}}\right| \leq \frac{|G(A_t, \sigma_t)_b|k}{\beta_t \gamma_t} \leq \frac{c_{\mathcal{G}}}{\beta_t \gamma_t} \leq \frac{c_{\mathcal{G}}}{u_t} = \frac{1 - \alpha}{8} \frac{1}{(\min\{q_{t\tilde{I}_t}, 1 - q_{t\tilde{I}_t}\})^{1-\alpha}}, \tag{61}
$$

where the third inequality follows from $\gamma_t \geq u_t/\beta_t$ in (11) and the last equality follows from the defintition of $u_t$ in (17). Hence, from Lemma 14 the LHS of Assumption (ii) is bounded as

$$
\begin{aligned}
\mathbb{E}_t\big[\langle \widehat{y}_t, q_t - q_{t+1}\rangle - \beta_t D_{(-H_\alpha)}(q_{t+1}, q_t)\big] &= \beta_t \mathbb{E}_t\left[\left\langle \frac{\widehat{y}_t}{\beta_t}, q_t - q_{t+1}\right\rangle - D_{(-H_\alpha)}(q_{t+1}, q_t)\right] \\
&\leq \mathbb{E}_t\left[\frac{4}{\beta_t(1-\alpha)}\left(\sum_{i \neq \tilde{I}_t} q_{ti}^{2-\alpha}\widehat{y}_{ti}^2 + (\min\{q_{t\tilde{I}_t}, 1 - q_{t\tilde{I}_t}\})^{2-\alpha}\widehat{y}_{t\tilde{I}_t}^2\right)\right] \\
&= \frac{4}{\beta_t(1-\alpha)}\left(\sum_{i \neq \tilde{I}_t} q_{ti}^{2-\alpha}\mathbb{E}_t\big[\widehat{y}_{ti}^2\big] + q_{t*}^{2-\alpha}\mathbb{E}_t\big[\widehat{y}_{t\tilde{I}_t}^2\big]\right).
\end{aligned} \tag{62}
$$

Since the variance of $\widehat{y}_t$ is bounded from above as

$$
\mathbb{E}_t\big[\widehat{y}_{ti}^2\big] = \sum_{a=1}^{k} p_{ta}\frac{G(a, \sigma_t)_i^2}{p_{ta}^2} \leq \sum_{a=1}^{k}\frac{k\|G\|_\infty^2}{\gamma_t} = \frac{c_{\mathcal{G}}^2}{\gamma_t} \tag{63}
$$

for any $i \in [k]$, the LHS of Assumption (ii) is further bounded as

$$
\mathbb{E}_t[\langle \widehat{y}_t, q_t - q_{t+1}\rangle - \beta_t D_{\psi_t}(q_{t+1}, q_t)] \leq \frac{4c_{\mathcal{G}}^2}{\beta_t \gamma_t(1-\alpha)}\left(\sum_{i \neq \tilde{I}_t} q_{ti}^{2-\alpha} + q_{t*}^{2-\alpha}\right) = \frac{z_t}{\beta_t \gamma_t} \leq \frac{z_t}{\beta_t \gamma_t'}, \tag{64}
$$

which implies that Assumption (ii) in (12) is satisfied.

Next, we will prove $h_{t+1} \lesssim h_t$ of Assumption (iii) in (12). To prove this, we will check the conditions (58) and (59) in Lemma 15. For any $a \in [k]$,

$$
|\widehat{y}_{ta}| \leq \frac{\|G\|_\infty}{p_{tA_t}} \leq \frac{k\|G\|_\infty}{\gamma_t} \leq \frac{c_{\mathcal{G}}\beta_t}{u_t} \leq \frac{1-\alpha}{8}\frac{\beta_t}{q_{t*}^{1-\alpha}} \leq \frac{1 - (\sqrt{2})^{\alpha-1}}{2}\frac{\beta_t}{q_{t*}^{1-\alpha}}, \tag{65}
$$

where the second inequality follows from $p_{ta} \geq \gamma_t/k$, the third inequality from $\gamma_t \geq u_t/\beta_t$, and the last inequality from the fact that $(1-x)/4 \leq 1 - (\sqrt{2})^{x-1}$ for $x \in [0, 1]$. Thus, the condition (58) is satisfied.

We next check the condition (59). Recalling $q_{t*} = \min\{q_{t\tilde{I}_t}, 1 - q_{t\tilde{I}_t}\}$, the parameters $z_t$ and $u_t$ satisfy

$$\sqrt{z_t} = \frac{2c_{\mathcal{G}}}{\sqrt{1-\alpha}}\sqrt{\sum_{i\neq\tilde{I}_t} q_{ti}^{2-\alpha} + q_{t*}^{2-\alpha}} \leq \frac{2\sqrt{k}c_{\mathcal{G}}}{\sqrt{1-\alpha}}q_{t*}^{1-\frac{1}{2}\alpha}, \quad u_t = \frac{8c_{\mathcal{G}}}{1-\alpha}q_{t*}^{1-\alpha}, \tag{66}$$

where the inequality follows from $q_{ti} \leq q_{t*}$ for $i \neq \tilde{I}_t$. The penalty component $h_t$ is lower-bounded as

$$h_t = H_\alpha(q_t) = \frac{1}{\alpha}\sum_{i=1}^{k}(q_{ti}^\alpha - q_{ti}) \geq \frac{1 - (1/2)^{1-\alpha}}{\alpha}q_{t*}^\alpha \geq \frac{1-\alpha}{4\alpha}q_{t*}^\alpha, \tag{67}$$

where the last inequality in (67) follows from $1 - (1/2)^{1-x} \geq (1-x)/4$ for $x \leq 0$, and the first inequality can be proven as follows: when $q_{t\tilde{I}_t} \leq 1/2$, it holds that $\sum_{i=1}^{k}(q_{ti}^\alpha - q_{ti}) \geq q_{t\tilde{I}_t}^\alpha - q_{t\tilde{I}_t} = q_{t\tilde{I}_t}^\alpha(1 - q_{t\tilde{I}_t}^{1-\alpha}) \geq q_{t\tilde{I}_t}^\alpha(1 - (1/2)^{1-\alpha}) = q_{t*}^\alpha(1 - (1/2)^{1-\alpha})$, and when $q_{t\tilde{I}_t} > 1/2$, it holds that $\sum_{i=1}^{k}(q_{ti}^\alpha - q_{ti}) \geq \sum_{i=1}^{k}q_{ti}^\alpha - 1 \geq \sum_{i\neq\tilde{I}_t}q_{ti}^\alpha + (1/2)^\alpha - 1 \geq (\sum_{i\neq\tilde{I}_t}q_{ti})^\alpha + (1/2)^\alpha - 1 = (1 - q_{t\tilde{I}_t})^\alpha + (1/2)^\alpha - 1 = q_{t*}^\alpha + (1/2)^\alpha - 1 \geq q_{t*}^\alpha(1 - (1/2)^{1-\alpha})$. Using the bounds on $z_t$, $u_t$, and $h_t$ in (66) and (67), we have

$$
\begin{aligned}
\beta_{t+1} - \beta_t &= \frac{1}{\tilde{h}_{t+1}}\left(2\sqrt{\frac{z_t}{\beta_t}} + \frac{u_t}{\beta_t}\right) = \frac{2}{h_t}\sqrt{\frac{z_t}{\beta_t}} + \frac{1}{h_t}\frac{u_t}{\beta_t} \\
&\leq \frac{16\alpha c_{\mathcal{G}}\sqrt{k}}{\sqrt{\beta_1}(1-\alpha)^{3/2}}q_{t*}^{1-\frac{3}{2}\alpha} + \frac{32\alpha c_{\mathcal{G}}}{\sqrt{\beta_1}(1-\alpha)^2}q_{t*}^{1-2\alpha} \\
&\leq \alpha\bar{\beta}q_{t*}^{1-\frac{3}{2}\alpha} + \alpha\bar{\beta}q_{t*}^{1-2\alpha} \\
&\leq 2(1-\bar{\alpha})\bar{\beta}q_{t*}^{\bar{\alpha}-\alpha} \leq 2\frac{1 - (\sqrt{2})^{\bar{\alpha}-1}}{\sqrt{2}}\bar{\beta}q_{t*}^{\bar{\alpha}-\alpha},
\end{aligned}
\tag{68}
$$

where the first inequality follows from (66), (67), and the fact that $\beta_t \geq \beta_1 \geq 1$, the second inequality from the definition of $\bar{\beta}$ in (17), the third inequality from $\min\{1 - \frac{3}{2}\alpha, 1 - 2\alpha\} \geq \bar{\alpha} - \alpha$ since $\bar{\alpha} = 1 - \alpha$, and the last inequality from $1 - x \leq (1 - (\sqrt{2})^{x-1})/\sqrt{2}$ for $x \leq 1$. Therefore, the condition (59) is satisfied. Hence, from Lemma 15, we have $h_{t+1} = H_\alpha(q_{t+1}) \leq 2H_\alpha(q_t) = 2h_t$, which implies that Assumption (iii) in (12) is satisfied.

Finally, we check the assumption (14) in Theorem 7. We first consider the first inequality in (14). From the definition of $z_t$ and the fact that $q_{ti} \leq q_{t\tilde{I}_t}$ for $i \neq \tilde{I}_t$, the stability component $z_t$ is bounded as

$$
\begin{aligned}
z_t &= \frac{4c_{\mathcal{G}}^2}{1-\alpha}\left\{\sum_{i\neq\tilde{I}_t}q_{ti}^{2-\alpha} + \left(\min\{q_{t\tilde{I}_t}, 1 - q_{t\tilde{I}_t}\}\right)^{2-\alpha}\right\} \\
&\leq \frac{4c_{\mathcal{G}}^2}{1-\alpha}\left\{\sum_{i\neq\tilde{I}_t}q_{ti}^{2-\alpha} + \left(\sum_{i\neq\tilde{I}_t}q_{ti}\right)^{2-\alpha}\right\} \\
&\leq \frac{8c_{\mathcal{G}}^2}{1-\alpha}\left(\sum_{i\neq\tilde{I}_t}q_{ti}\right)^{2-\alpha} \leq \frac{8c_{\mathcal{G}}^2}{1-\alpha}\left(\sum_{i\neq a^*}q_{ti}\right)^{2-\alpha} = \frac{8c_{\mathcal{G}}^2}{1-\alpha}(1 - q_{ta^*})^{2-\alpha},
\end{aligned}
\tag{69}
$$

where the second inequality holds from the inequality $x^a + y^a \leq (x+y)^a$ for $x, y \geq 0$ and $a \geq 1$, and the third inequality from $q_{ti} \leq q_{t\tilde{I}_t}$ for $i \neq \tilde{I}_t$. From Lemma 13, we also obtain that

$$h_t = H_\alpha(q_t) \leq \frac{1}{\alpha}(k-1)^{1-\alpha}(1 - q_{ta^*})^\alpha. \tag{70}$$

Hence, combining (69) and (70), we obtain

$$z_t h_t \leq \frac{8c_{\mathcal{G}}^2}{1-\alpha}(1 - q_{ta^*})^{2-\alpha}\cdot\frac{1}{\alpha}(k-1)^{1-\alpha}(1 - q_{ta^*})^\alpha = \underbrace{\frac{8c_{\mathcal{G}}^2(k-1)^{1-\alpha}}{\alpha(1-\alpha)}}_{=\rho_1}(1 - q_{ta^*})^2. \tag{71}$$

We next consider the second inequality in (14). We can bound $u_t$ from above as

$$u_t = \frac{8c_{\mathcal{G}}}{1-\alpha}\left(\min\{q_{t\tilde{I}_t}, 1 - q_{t\tilde{I}_t}\}\right)^{1-\alpha} \leq \frac{8c_{\mathcal{G}}}{1-\alpha}\left(\sum_{i \neq \tilde{I}_t} q_{ti}\right)^{1-\alpha}$$

$$\leq \frac{8c_{\mathcal{G}}}{1-\alpha}\left(\sum_{i \neq a^*} q_{ti}\right)^{1-\alpha} = \frac{8c_{\mathcal{G}}}{1-\alpha}(1 - q_{ta^*})^{1-\alpha}, \tag{72}$$

where the second inequality follows from $q_{t\tilde{I}_t} \geq q_{ti}$ for all $i \in [k]$. Hence, combining the last two inequality and (70),

$$u_t h_t \leq \underbrace{\frac{4c_{\mathcal{G}}(k-1)^{1-\alpha}}{\alpha(1-\alpha)}}_{=\rho_2}(1 - q_{ta^*}). \tag{73}$$

Hence, the assumption (14) is satisfied with above $\rho_1$ and $\rho_2$, and thus we have completed the proof. $\square$

# F Proof for graph bandits (Theorem 10, Section 6)

This section provides the missing detail of Section 6.

## F.1 Fractional domination number

Before introducing the fractional domination number, we define the domination number $\tilde{\delta} \leq \delta$. A *dominating set* $D \subseteq V$ is a set of vertices such that $V \subseteq \bigcup_{i \in D} N^{\mathsf{out}}(i)$. The *domination number* $\tilde{\delta}(G)$ of graph $G$ is the size of the smallest dominating set. From the definition, the domination number $\tilde{\delta}$ can also be written as the optimal value of the following optimization problem:

$$\text{minimize} \sum_{i \in V} x_i \quad \text{subject to} \quad \sum_{i \in N^{\mathsf{in}}(j)} x_i \geq 1 \ \forall j \in V, \ x_i \in \{0, 1\} \ \forall i \in V, \tag{74}$$

where $x_i \in \{0, 1\}$ a binary variable indicating whether vertex $i$ is in the dominating set ($x_i = 1$) or not ($x_i = 0$).

Then, one can see that the fractional domination number $\delta^*$ is defined as the optimal value of the following optimization problem, in which the variables $(x_i)_{i \in V}$ are allowed to take values in $[0, 1]$ instead of $\{0, 1\}$:

$$\text{minimize} \sum_{i \in V} x_i \quad \text{subject to} \quad \sum_{i \in N^{\mathsf{in}}(j)} x_i \geq 1 \ \forall j \in V, \ 0 \leq x_i \leq 1 \ \forall i \in V, \tag{75}$$

which is the linear program provided in (19). From the definitions, the fractional domination number is less than or equal to the domination number, $\delta^* \leq \tilde{\delta}$. Another advantage of using $\delta^*$ instead of $\tilde{\delta}$ is that the fractional domination number $\delta^*$ can be computed in polynomial time, while the computation of the domination number $\tilde{\delta}$ is NP-hard. See [13] for more benefits of using the fractional version of the (weak) domination number.

## F.2 Proof of Theorem 10

Here, we provide the proof of Theorem 10.

*Proof.* It suffices to prove that assumptions in Theorem 7 are satisfied. We first vertify Assumptions (i)–(iii) in (12). We start by checking Assumption (i). The regret is bounded as

$$\mathsf{Reg}_T = \mathbb{E}\left[\sum_{t=1}^T \ell_t(A_t) - \sum_{t=1}^T \ell_t(a^*)\right] = \mathbb{E}\left[\sum_{t=1}^T \langle \ell_t, p_t - e_{a^*}\rangle\right] = \mathbb{E}\left[\sum_{t=1}^T \langle \ell_t, q_t - e_{a^*}\rangle + \sum_{t=1}^T \langle \ell_t, p_t - q_t\rangle\right]$$

$$= \mathbb{E}\left[\sum_{t=1}^T \langle \ell_t, q_t - e_{a^*}\rangle + \sum_{t=1}^T \gamma_t\langle \ell_t, q_t - u\rangle\right] \leq \mathbb{E}\left[\sum_{t=1}^T \langle \widehat{\ell}_t, q_t - e_{a^*}\rangle + \sum_{t=1}^T \gamma_t\right], \tag{76}$$

where the third equality follows from the definition of $\gamma_t$. This implies that Assumption (i) is indeed satisfied.

We next check Assumption (ii) in (12). Now, recalling the definition of the fractional domination number and the optimal value $x^*$ of (19), and $u_i = x_i^* / \sum_{j \in V} x_j^*$, we have

$$\sum_{j \in N^{\text{in}}(i)} u_j = \frac{\sum_{j \in N^{\text{in}}(i)} x_j^*}{\sum_{i \in V} x_i^*} \geq \frac{1}{\sum_{i \in V} x_i^*} = \frac{1}{\delta^*} , \tag{77}$$

where the inequality follows from the first constraint in (19). Hence, combining this with the definition of $p_t = (1 - \gamma_t)q_t + \gamma_t u$, we can lower-bound $P_{ti}$ as

$$P_{ti} = \sum_{j \in N^{\text{in}}(i)} p_{tj} \geq \gamma_t \sum_{j \in N^{\text{in}}(i)} u_j \geq \frac{\gamma_t}{\delta^*} \quad \text{for all } i \in V . \tag{78}$$

This lower bound yields that for any $i \in V$

$$\left| \frac{\widehat{\ell}_{ti}}{\beta_t} \right| \leq \frac{\ell_{ti}}{\beta_t P_{ti}} \leq \frac{\delta^*}{\beta_t \gamma_t} \leq \frac{\delta^*}{u_t} = \frac{1 - \alpha}{8} \frac{1}{\left( \min\{q_{t\tilde{I}_t}, 1 - q_{t\tilde{I}_t}\} \right)^{1-\alpha}} , \tag{79}$$

where the second inequality follows from (78) and the third inequality from $\gamma_t \geq u_t / \beta_t$ in (11). Hence, from Lemma 14 we obtain

$$\mathbb{E}_t \left[ \left\langle \widehat{\ell}_t, q_t - q_{t+1} \right\rangle - \beta_t D_{(-H_\alpha)}(q_{t+1}, q_t) \right] = \beta_t \mathbb{E}_t \left[ \left\langle \frac{\widehat{\ell}_t}{\beta_t}, q_t - q_{t+1} \right\rangle - D_{(-H_\alpha)}(q_{t+1}, q_t) \right]$$

$$\leq \mathbb{E}_t \left[ \frac{4}{\beta_t(1 - \alpha)} \left( \sum_{i \in V \setminus \{\tilde{I}_t\}} q_{ti}^{2-\alpha} \widehat{\ell}_{ti}^2 + \left( \min\{q_{t\tilde{I}_t}, 1 - q_{t\tilde{I}_t}\} \right)^{2-\alpha} \widehat{\ell}_{t\tilde{I}_t}^2 \right) \right]$$

$$= \frac{4}{\beta_t(1 - \alpha)} \left( \sum_{i \in V \setminus \{\tilde{I}_t\}} q_{ti}^{2-\alpha} \mathbb{E}_t \left[ \widehat{\ell}_{ti}^2 \right] + q_{t*}^{2-\alpha} \mathbb{E}_t \left[ \widehat{\ell}_{t\tilde{I}_t}^2 \right] \right) . \tag{80}$$

Then, by using the lower bound of $P_t$ in (78), for any $i \in V$ the variance of the loss estimator $\widehat{\ell}_{ti}$ is bounded as

$$\mathbb{E}_t \left[ \widehat{\ell}_{ti}^2 \right] = \sum_{j=1}^{k} p_{tj} \frac{\ell_{ti}^2}{P_{ti}^2} \mathbb{1}\left[ i \in N^{\text{out}}(j) \right] = \frac{\ell_{ti}^2}{P_{ti}^2} \sum_{j \in V : i \in N^{\text{out}}(j)} p_{tj} = \frac{\ell_{ti}^2}{P_{ti}} \leq \frac{\delta^*}{\gamma_t} . \tag{81}$$

Hence, combining (80) with (81), we obtain

$$\mathbb{E}_t[\langle \widehat{y}_t, q_t - q_{t+1} \rangle - \beta_t D_{\psi_t}(q_{t+1}, q_t)] \leq \frac{4\delta^*}{\beta_t \gamma_t (1 - \alpha)} \left( \sum_{i \in V \setminus \{\tilde{I}_t\}} q_{ti}^{2-\alpha} + q_{t*}^{2-\alpha} \right) = \frac{z_t}{\beta_t \gamma_t} \leq \frac{z_t}{\beta_t \gamma_t'} , \tag{82}$$

which implies that Assumption (ii) in (12) is satisfied.

Next, we will prove $h_{t+1} \lesssim h_t$ of Assumption (iii) in (12). To prove this, we will check the conditions (58) and (59) in Lemma 15. For any $i \in V$,

$$|\widehat{\ell}_{ti}| \leq \frac{1}{P_{ti}} \leq \frac{\delta^*}{\gamma_t} \leq \frac{\delta^* \beta_t}{u_t} = \frac{1 - \alpha}{8} \frac{\beta_t}{q_{t*}^{1-\alpha}} \leq \frac{1 - (\sqrt{2})^{\alpha-1}}{2} \frac{\beta_t}{q_{t*}^{1-\alpha}} , \tag{83}$$

where the second inequality follows from (78), the third inequality from $\gamma_t \geq u_t / \beta_t$, and the last inequality from the fact that $(1 - x)/4 \leq 1 - (\sqrt{2})^{x-1}$ for $x \in [0, 1]$. Thus, the condition (58) is satisfied.

We next check the condition (59). Recalling $q_{t*} = \min\{q_{t\tilde{I}_t}, 1 - q_{t\tilde{I}_t}\}$, we observe that the parameters $z_t$ and $u_t$ satisfy

$$\sqrt{z_t} = \sqrt{\frac{4\delta^*}{1 - \alpha} \left( \sum_{i \in V \setminus \{\tilde{I}_t\}} q_{ti}^{2-\alpha} + q_{t*}^{2-\alpha} \right)} \leq \frac{2\sqrt{k\delta^*}}{\sqrt{1 - \alpha}} q_{t*}^{1 - \frac{1}{2}\alpha} , \quad u_t = \frac{8\delta^*}{1 - \alpha} q_{t*}^{1-\alpha} , \tag{84}$$

where the last inequality follows from $q_{ti} \leq q_{t*}$ for $i \neq \tilde{I}_t$. We can also lower-bound $h_t$ as

$$h_t = H_\alpha(q_t) = \frac{1}{\alpha} \sum_{i=1}^{k} (q_{ti}^\alpha - q_{ti}) \geq \frac{1 - (1/2)^{1-\alpha}}{\alpha} q_{t*}^\alpha \geq \frac{1 - \alpha}{4\alpha} q_{t*}^\alpha, \tag{85}$$

which can be proven in the same manner as in (67). Hence, using the upper bounds on $z_t$, $u_t$, and $h_t$ in (84) and (85), we have

$$\beta_{t+1} - \beta_t = \frac{1}{\hat{h}_{t+1}} \left( 2\sqrt{\frac{z_t}{\beta_t}} + \frac{u_t}{\beta_t} \right) = \frac{2}{h_t} \sqrt{\frac{z_t}{\beta_t}} + \frac{1}{h_t} \frac{u_t}{\beta_t}$$

$$\leq \frac{16\alpha\sqrt{k\delta^*}}{\sqrt{\beta_1}(1-\alpha)^{3/2}} q_{t*}^{1-\frac{3}{2}\alpha} + \frac{32\alpha\delta^*}{\sqrt{\beta_1}(1-\alpha)^2} q_{t*}^{1-2\alpha}$$

$$\leq \alpha\bar{\beta} q_{t*}^{1-\frac{3}{2}\alpha} + \alpha\bar{\beta} q_{t*}^{1-2\alpha}$$

$$\leq 2(1-\bar{\alpha})\bar{\beta} q_{t*}^{\bar{\alpha}-\alpha} \leq 2\frac{1 - (\sqrt{2})^{\bar{\alpha}-1}}{\sqrt{2}} \bar{\beta} q_{t*}^{\bar{\alpha}-\alpha}, \tag{86}$$

where the first inequality follows from (84), (85), and $\beta_t \geq \beta_1 \geq 1$, the second inequality from the definition of $\bar{\beta}$, the third inequality from $\min\{1 - \frac{3}{2}\alpha, 1 - 2\alpha\} \geq \bar{\alpha} - \alpha$ since $\bar{\alpha} = 1 - \alpha$, and the last inequality from $1 - x \leq (1 - (\sqrt{2})^{x-1})/\sqrt{2}$ for $x \leq 1$. Thus the condition (59) is satisfied. Therefore, from Lemma 15, we have $h_{t+1} = H_\alpha(q_{t+1}) \leq 2H_\alpha(q_t) = 2h_t$, which implies that Assumption (iii) in (12) is satisfied.

Finally, we check the assumption (14) in Theorem 7. We first consider the first inequality in (14). From the definition of $z_t$ and the fact that $q_{ti} \leq q_{t\tilde{I}_t}$ for $i \neq \tilde{I}_t$, we get

$$z_t = \frac{4\delta^*}{1-\alpha} \left\{ \sum_{i \in V \setminus \{\tilde{I}_t\}} q_{ti}^{2-\alpha} + \left( \min\{q_{t\tilde{I}_t}, 1 - q_{t\tilde{I}_t}\} \right)^{2-\alpha} \right\}$$

$$\leq \frac{4\delta^*}{1-\alpha} \left\{ \sum_{i \in V \setminus \{\tilde{I}_t\}} q_{ti}^{2-\alpha} + \left( \sum_{i \neq \tilde{I}_t} q_{ti} \right)^{2-\alpha} \right\}$$

$$\leq \frac{8\delta^*}{1-\alpha} \left( \sum_{i \in V \setminus \{\tilde{I}_t\}} q_{ti} \right)^{2-\alpha} \leq \frac{8\delta^*}{1-\alpha} \left( \sum_{i \neq a^*} q_{ti} \right)^{2-\alpha} = \frac{8\delta^*}{1-\alpha} (1 - q_{ta^*})^{2-\alpha}, \tag{87}$$

where the second inequality holds from the inequality $x^a + y^a \leq (x + y)^a$ for $x, y \geq 0$ and $a \geq 1$, and the third inequality from $q_{ti} \leq q_{t\tilde{I}_t}$. Hence, combining (87) and the upper bound on $h_t$ in (70), we obtain

$$z_t h_t \leq \frac{8\delta^*}{1-\alpha} (1 - q_{ta^*})^{2-\alpha} \cdot \frac{1}{\alpha} (k-1)^{1-\alpha} (1 - q_{ta^*})^\alpha = \underbrace{\frac{8\delta^*(k-1)^{1-\alpha}}{\alpha(1-\alpha)}}_{=\rho_1} (1 - q_{ta^*})^2. \tag{88}$$

We next consider the second inequality in (14). We can bound $u_t$ from above as

$$u_t = \frac{8\delta^*}{1-\alpha} \left( \min\{q_{t\tilde{I}_t}, 1 - q_{t\tilde{I}_t}\} \right)^{1-\alpha} \leq \frac{8\delta^*}{1-\alpha} \left( \sum_{i \neq \tilde{I}_t} q_{ti} \right)^{1-\alpha}$$

$$\leq \frac{8\delta^*}{1-\alpha} \left( \sum_{i \neq a^*} q_{ti} \right)^{1-\alpha} = \frac{8\delta^*}{1-\alpha} (1 - q_{ta^*})^{1-\alpha}, \tag{89}$$

where the second inequality follows from $q_{t\tilde{I}_t} \geq q_{ti}$ for all $i \neq \tilde{I}_t$. Hence, combining the last inequality with (70),

$$u_t h_t \leq \underbrace{\frac{4\delta^*(k-1)^{1-\alpha}}{\alpha(1-\alpha)}}_{=\rho_2} (1 - q_{ta^*}). \tag{90}$$

Hence, the assumption (14) is satisfied with above $\rho_1$ and $\rho_2$, and thus we have completed the proof. $\qquad\square$

### F.3 Technical challenges to derive best-of-both-worlds bounds depending on (fractional) weak domination number

Here, we discuss the technical challenges of making our upper bound in Theorem 10 depend on the weak domination number $\delta$ instead of the fracional domination number $\delta^*$ or the weak fractional domination number $\tilde{\delta}^* \leq \delta$.

First, we need to use Tsallis entropy to derive a regret upper bound with a stochastic bound of $\log T$. While we can prove a BOBW bound if we use the Shannon entropy regularizer [25], the bound in the stochastic regime is $O((\log T)^2)$, which is not desirable. Hence, a possible approach is to use the log-barrier regularizer or the Tsallis entropy. The log-barrier regularizer has a penalty term of $\Omega(k)$ due to the strength of its regularization, and the regret upper bound in the final adversarial regime is $\Omega(k^{1/3})$, which can be much larger than $\delta^{1/3}$. Therefore, the most hopeful solution would be to use Tsallis entropy with an appropriate exponent $\alpha \simeq 1$, where we note that the Tsallis entropy with $\alpha \to 1$ corresponds to the Shannon entropy.

Recalling the definition of the weak domination number in Section 6, we can see that the weak dominating set dominates only vertices without self-loop $U = \{i \in V : i \notin N^{\text{out}}(i)\}$. Thus, to achieve a BOBW bound that depends on the weak domination number, vertices with self-loop and those without self-loop should be treated separately by decomposing the stability term as follows:

$$\langle \widehat{\ell}_t, q_t - q_{t+1} \rangle - \beta_t\, D_{(-H_\alpha)}(q_{t+1}, q_t)$$
$$= \sum_{i \in U} \left( \widehat{\ell}_{ti}(q_{ti} - q_{t+1,i}) - \beta_t\, d(q_{t+1,i}, q_{t,i}) \right) + \sum_{i \in V \setminus U} \left( \widehat{\ell}_{ti}(q_{ti} - q_{t+1,i}) - \beta_t\, d(q_{t+1,i}, q_{t,i}) \right),$$

where $d(p, q)$ is the Bregman divergence induced by the real-valued convex function $x \mapsto -\frac{1}{\alpha}(x^\alpha - x)$. However, if we use this approach, we cannot use Lemma 14, which is useful to prove an upper bound with $(1 - q_{ta^*})$ (see (14)). This is because this lemma exploits the fact that $q$ and $r$ are probability vectors. This prevents us from deriving an upper bound with an $O(\log T)$ stochastic bound depending on the weak domination number.

## G  Case study (3): Multi-armed bandits with paid observations

### G.1  Problem setting and existing approach

Multi-armed bandits with paid observations, which is first investigated by Seldin et al. [53], is a variant of the multi-armed bandit problem. At each round $t \in [T]$, the environment determines a loss vector $\ell_t \colon \mathcal{A} = [k] \to [0, 1]$ and the learner observes cost vector $c_t \in \mathbb{R}^k_{\geq 0}$. Then, the learner selects an action $A_t \in [k]$ and chooses a set of actions $S_t \subseteq [k]$, for which we can observe losses. Then the learner suffers a loss of $\ell_{tA_t} + \sum_{i \in S_t} c_{ti}$ and observes a set of losses $\{\ell_{ti} : i \in S_t\}$. The goal of the learner is to minimize the sum of the standard regret and the observation costs given by

$$\mathsf{Reg}^{\mathsf{cost}}_T = \mathsf{Reg}_T + \mathbb{E}\left[ \sum_{t=1}^{T} \sum_{i \in S_t} c_{ti} \right]. \tag{91}$$

We next provide an existing approach to determine the set $S_t$ and to estimate losses, which are given by Seldin et al. [53]. To determine $S_t$, we prepare a vector $r_t \in [0, 1]^k$. For this $r_t$, we then sample $b_{ti} \sim \mathrm{Ber}(r_{ti})$ for each $i \in [k]$, and use this to construct the set of actions $S_t = \{i \in [k] : b_{ti} = 1\}$. We use the loss estimator defined by

$$\widehat{\ell}_{ti} = \frac{\ell_{ti}}{r_{ti}} \mathbb{1}[i \in S_t], \tag{92}$$

which is indeed unbiased, $\mathbb{E}_{S_t}[\widehat{\ell}_{ti}] = \ell_{ti}$.

In the following, we assume that the cost is the same for each arm at each time, $c_{ti} = c \geq 0$. Accordingly, we let $r_{ti} = r_t \in [0, 1]$ for each $i \in [k]$, where we abuse the notation. Analyzing a case

---

**Algorithm 2:** Best-of-both-worlds algorithm based on FTRL with SPB-matching learning rate and Tsallis entropy in multi-armed bandits with paid observations

---

1   **input:** action set $\mathcal{A} = [k]$, exponent of Tsallis entropy $\alpha$, $\beta_1$, $\bar{\beta}$

2   **for** $t = 1, 2, \ldots$ **do**

3      Compute $q_t \in \mathcal{P}_k$ by (10) with a loss estimator $\widehat{\ell}_t$ in (92).

4      Set $h_t = H_\alpha(q_t)$ and $z_t, u_t \geq 0$ in (94).

5      Compute action selection probability $p_t = q_t$ without forced exploration.

6      For $r_t \in [0, 1]$ in (93), sample $b_{ti} \sim \mathrm{Ber}(r_t)$ for each $i \in \mathcal{A}$ and let $S_t = \{i \in [k] \colon b_{ti} = 1\}$.

7      Choose $A_t \in [k]$ so that $\Pr[A_t = i \mid p_t] = p_{ti}$.

8      Observes the set of losses $\{\ell_{ti} \colon i \in S_t\}$ and suffers a loss of $\ell_{tA_t} + \sum_{i \in S_t} c_{ti}$.

9      Compute loss estimator $\widehat{\ell}_t$ based on $r_t$ and $S_t$.

10     Compute $\beta_{t+1}$ by Rule 2 of SPB-matching in (6) with $\widehat{h}_{t+1} = h_t$.

---

where each action has a different cost and deriving an upper bound that depends on the cost of each action is difficult. This is essentially due to the same reason as the problem in graph bandits with self-loops, where the regret upper bound depends on the domination number (see Appendix F.3).

The setting of multi-armed bandits with paid observations is not directly reducible to the general online learning framework defined in Section 2. However, the parameter $r_t$ plays the same role as the forced exploration parameter $\gamma_t$ in partial monitoring and graph bandits, and thus their regret upper bounds have a similar structure. Roughly speaking, we will see in the regret analysis of multi-armed bandits with paid observations with cost $c \geq 0$ can be regarded as the general online learning setup with the exploration rate of $\gamma_t \simeq ckr_t$.

## G.2   Algorithm

We use FTRL provided in (10) and (11) as for graph bandits and partial monitoring with no forced exploration, that is, $p_t = q_t$. Here we recall that $p_t \in \mathcal{P}_k$ is the action selection probability at round $t$ and $q_t \in \mathcal{P}_k$ is the output of FTRL at round $t$. We use $r_t \in [0, 1]$ given by

$$r_t = \sqrt{\frac{z_t}{\beta_t}} + \frac{u_t}{\beta_t}, \tag{93}$$

which plays a role of exploration rate $\gamma_t$. We will choose $\beta_1$ so that $r_t \leq 1/2$. Next we specify the parameters in (11). For $\tilde{I}_t \in \arg\max_{i \in [k]} q_{ti}$ and $q_{t*} = \min\{q_{t\tilde{I}_t}, 1 - q_{t\tilde{I}_t}\}$, we use

$$\beta_1 \geq \frac{64 \max\{c, 1\} k}{1 - \alpha} \,, \quad \bar{\beta} = \frac{32 k \sqrt{c}}{(1 - \alpha)^2 \sqrt{\beta_1}} \,, \quad z_t = \frac{4ck}{1 - \alpha} \left( \sum_{i \neq \tilde{I}_t} q_{ti}^{2-\alpha} + q_{t*}^{2-\alpha} \right) , \quad u_t = \frac{8 \max\{c, 1\}}{1 - \alpha} q_{t*}^{1-\alpha} \,. \tag{94}$$

Note that $z_{\max} = \frac{4c}{1-\alpha}$, $u_{\max} = \frac{8 \max\{c,1\}}{1-\alpha}$, and $h_{\max} = h_1 = \frac{1}{\alpha} k^{1-\alpha}$, and the above $\beta_1$ implies $r_t \leq 1/2$. To follow the analysis in the general online learning framework, we also let $r'_t = \sqrt{z_t/\beta_t}$ and $\gamma'_t = ckr'_t \leq \gamma_t$. To make the algorithm clear, we provide the full description of our algorithm in Algorithm 2.

## G.3   Regret analysis

We can prove the following.

**Theorem 16.** *In multi-armed bandits with paid observations, for any $\alpha \in (0, 1)$, Algorithm 2 satisfies the assumptions of Theorem 7 with $\gamma_t = ckr_t$, $\rho_1 = \Theta\left( \frac{ck^{2-\alpha}}{\alpha(1-\alpha)} \right)$, and $\rho_2 = \Theta\left( \frac{\max\{c,1\} k^{1-\alpha}}{\alpha(1-\alpha)} \right)$, where the regret $\mathrm{Reg}_T$ in the statement is repalced with $\mathrm{Reg}_T^{\mathrm{cost}}$.*

Note that here we are abusing the statement of Theorem 7 since Theorem 7 is for the general online learning framework given in Section 2 but the multi-armed bandits with paid observation is not a special case of the general online learning framework. Still, if we set the exploration rate $\gamma_t$ to

$\gamma_t = ckr_t$, then the minimization of the regret with costs, $\text{Reg}_T^{\text{cost}}$, in multi-armed bandits with paid observation under paid cost $c$ and parameter $r_t$ can be seen as the minimization of the regret $\text{Reg}_T$ in the general online learning framework with exploration rate $\gamma_t = ckr_t$. A formal proof of the theorem for multi-armed bandits with paid observations corresponding to Theorem 7 follows the same argument and we omit it.

Setting $\alpha = 1 - 1/(\log k)$ in the last theorem gives the following:

**Corollary 17.** *In multi-armed bandits with paid observation with $T \geq \tau$, Algorithm 2 with $\alpha = 1 - 1/(\log k)$ achieves*

$$
\text{Reg}_T^{\text{cost}} = \begin{cases} O\big((ck)^{1/3}T^{2/3}(\log k)^{1/3} + \sqrt{T\log k} + \kappa\big) & \text{in adversarial regime} \\[2ex] O\left(\dfrac{\max\{c,1\}k\log k}{\Delta_{\min}^2}\log\big(T\Delta_{\min}^3\big) + \left(\dfrac{C^2\max\{c,1\}k\log k}{\Delta_{\min}^2}\log\left(\dfrac{T\Delta_{\min}}{C}\right)\right)^{1/3} + \kappa'\right) \\[2ex] \qquad\qquad \text{in adversarial regime with a } (\Delta, C, T)\text{-self-bounding constraint .} \end{cases}
$$
$$\tag{95}$$

*Here, if we use $\beta_1 = 64\max\{c,1\}k/(1-\alpha)$, which satisfies (94), $\kappa = O(\max\{c,1\}k\log k + k^{3/2}(\log k)^{5/2})$ and $\kappa' = \kappa + O\big(((c\log k)^{1/3} + \sqrt{\max\{c,1\}\log k})(\frac{1}{\Delta_{\min}^3} + \frac{C}{\Delta_{\min}})^{2/3}\big)$.*

This regret upper bound is the first BOBW bounds in multi-armed bandits with paid observations. The upper bound in the adversarial regime becomes $O(\sqrt{T\log k})$ as $c \to 0$, as observed in [53]. The bound in the stochastic regime can also match the nearly optimal regret bound of $O(\log k \log T/\Delta_{\min})$ in the expert problem when $c \to 0$. To formally check this, it suffices to refine the analysis in Theorem 7 by analyzing $\rho_1$ and $\rho_2$ separately, which is unified into $\rho = \max\{\rho_1, \rho_2\}$ for simplicity of notation in the proof of Theorem 7.

*Proof of Theorem 16.* From the observation that the variable $r_t$ plays the same role as the exploration parameter $\gamma_t$, it suffices to prove that assumptions in Theorem 7 are satisfied. We first vertify Assumptions (i)–(iii) in (12). We start by checking Assumption (i). The regret with costs is bounded as

$$
\begin{aligned}
\text{Reg}_T^{\text{cost}} &= \mathbb{E}\left[\sum_{t=1}^T \ell_{tA_t} - \sum_{t=1}^T \ell_{ta^*} + \sum_{t=1}^T \sum_{i\in O_t} c_{ti}\right] = \mathbb{E}\left[\sum_{t=1}^T \langle \ell_t, p_t - e_{a^*}\rangle + \sum_{t=1}^T \langle r_t, c_t\rangle\right] \\
&= \mathbb{E}\left[\sum_{t=1}^T \langle \widehat{\ell}_t, p_t - e_{a^*}\rangle + \sum_{t=1}^T \langle r_t\mathbf{1}, c\mathbf{1}\rangle\right] \\
&= \mathbb{E}\left[\sum_{t=1}^T \langle \widehat{\ell}_t, p_t - e_{a^*}\rangle + kc\sum_{t=1}^T r_t\right],
\end{aligned}
$$
$$\tag{96}$$

where we recall that we are abusing the notation so that $r_{ti} = r_t \in [0,1]$. This implies that Assumption (i) is satisfied with $\gamma_t = ckr_t$.

We next check Assumption (ii) in (12). For any $i \in [k]$,

$$
\left|\frac{\widehat{\ell}_{ti}}{\beta_t}\right| \leq \frac{\ell_{ti}}{\beta_t r_{ti}} \leq \frac{1}{u_t} \leq \frac{1-\alpha}{8}\frac{1}{\big(\min\{q_{t\tilde{I}_t}, 1-q_{t\tilde{I}_t}\}\big)^{1-\alpha}},
$$
$$\tag{97}$$

where the first inequality follows from the definition of $\widehat{\ell}_t$, the second inequality from $r_t \geq u_t/\beta_t$, and the last inequality from the definition of $u_t$. Note that this is where $\max\{c,1\}$ in $u_t$ is used.

Hence, from Lemma 14 we obtain

$$\mathbb{E}_t\left[\left\langle \widehat{\ell}_t, q_t - q_{t+1}\right\rangle - \beta_t\, D_{(-H_\alpha)}(q_{t+1}, q_t)\right] = \beta_t\mathbb{E}_t\left[\left\langle \frac{\widehat{\ell}_t}{\beta_t}, q_t - q_{t+1}\right\rangle - D_{(-H_\alpha)}(q_{t+1}, q_t)\right]$$

$$\leq \mathbb{E}_t\left[\frac{4}{\beta_t(1-\alpha)}\left(\sum_{i\neq \tilde{I}_t} q_{ti}^{2-\alpha}\widehat{\ell}_{ti}^2 + \left(\min\{q_{t\tilde{I}_t}, 1-q_{t\tilde{I}_t}\}\right)^{2-\alpha}\widehat{\ell}_{t\tilde{I}_t}^2\right)\right]$$

$$= \frac{4}{\beta_t(1-\alpha)}\left(\sum_{i\neq \tilde{I}_t} q_{ti}^{2-\alpha}\mathbb{E}_t\left[\widehat{\ell}_{ti}^2\right] + q_{t*}^{2-\alpha}\mathbb{E}_t\left[\widehat{\ell}_{t\tilde{I}_t}^2\right]\right). \tag{98}$$

Now, for any $i \in [k]$ the variance of the loss estimator $\widehat{\ell}_{ti}$ is bounded as

$$\mathbb{E}_t\left[\widehat{\ell}_{ti}^2\right] = \mathbb{E}_t\left[\frac{\ell_{ti}^2}{r_t^2}\mathbb{1}[i \in S_t]\right] \leq \frac{1}{r_t}. \tag{99}$$

Hence, combining (98) with (99), we obtain

$$\mathbb{E}_t[\langle \widehat{y}_t, q_t - q_{t+1}\rangle - \beta_t D_{\psi_t}(q_{t+1}, q_t)]$$

$$\leq \frac{4}{\beta_t r_t(1-\alpha)}\left(\sum_{i\neq \tilde{I}_t} q_{ti}^{2-\alpha} + q_{t*}^{2-\alpha}\right) = \frac{4ck}{\beta_t\gamma_t(1-\alpha)}\left(\sum_{i\neq \tilde{I}_t} q_{ti}^{2-\alpha} + q_{t*}^{2-\alpha}\right) = \frac{z_t}{\beta_t\gamma_t} \leq \frac{z_t}{\beta_t\gamma_t'}, \tag{100}$$

where the first equality follows from $\gamma_t = ckr_t$. This implies that Assumption (ii) in (12) is satisfied.

Next, we will prove $h_{t+1} \lesssim h_t$ of Assumption (iii) in (12). To prove this, we will check the conditions (58) and (59) in Lemma 15. For any $i \in [k]$,

$$|\widehat{\ell}_{ti}| \leq \frac{1}{r_t} \leq \frac{\beta_t}{u_t} = \frac{1-\alpha}{8}\frac{\beta_t}{q_{t*}^{1-\alpha}} \leq \frac{1-(\sqrt{2})^{\alpha-1}}{2}\frac{\beta_t}{q_{t*}^{1-\alpha}}, \tag{101}$$

where the second inequality follows from $r_t \geq u_t/\beta_t$ and the last inequality from the fact that $(1-x)/4 \leq 1-(\sqrt{2})^{x-1}$ for $x \in [0,1]$. Thus, the condition (58) is satisfied.

We next check the condition (59). Recalling $q_{t*} = \min\{q_{t\tilde{I}_t}, 1-q_{t\tilde{I}_t}\}$, we observe that the parameters $z_t$ and $u_t$ satisfy

$$\sqrt{z_t} = \sqrt{\frac{4ck}{1-\alpha}\left(\sum_{i\neq \tilde{I}_t} q_{ti}^{2-\alpha} + q_{t*}^{2-\alpha}\right)} \leq \frac{2k\sqrt{c}}{\sqrt{1-\alpha}}q_{t*}^{1-\frac{1}{2}\alpha}, \qquad u_t = \frac{8\max\{c,1\}}{1-\alpha}q_{t*}^{1-\alpha}, \tag{102}$$

where the inequality follows from $q_{ti} \leq q_{t*}$ for $i \neq \tilde{I}_t$. We can also lower bound $h_t$ as

$$h_t = H_\alpha(q_t) = \frac{1}{\alpha}\sum_{i=1}^k (q_{ti}^\alpha - q_{ti}) \geq \frac{1-(1/2)^{1-\alpha}}{\alpha}q_{t*}^\alpha \geq \frac{1-\alpha}{4\alpha}q_{t*}^\alpha, \tag{103}$$

which can be proven by the same manner as in (67). Hence, using the upper bounds on $z_t$, $u_t$, and $h_t$ in (102) and (103), we have

$$\beta_{t+1} - \beta_t = \frac{1}{\widehat{h}_{t+1}}\left(2\sqrt{\frac{z_t}{\beta_t}} + \frac{u_t}{\beta_t}\right) = \frac{2}{h_t}\sqrt{\frac{z_t}{\beta_t}} + \frac{1}{h_t}\frac{u_t}{\beta_t}$$

$$\leq \frac{16\alpha\sqrt{kc}}{\sqrt{\beta_1}(1-\alpha)^{3/2}}q_{t*}^{1-\frac{3}{2}\alpha} + \frac{32\alpha\max\{c,1\}}{\sqrt{\beta_1}(1-\alpha)^2}q_{t*}^{1-2\alpha}$$

$$\leq \alpha\bar{\beta}q_{t*}^{1-\frac{3}{2}\alpha} + \alpha\bar{\beta}q_{t*}^{1-2\alpha}$$

$$\leq 2(1-\bar{\alpha})\bar{\beta}q_{t*}^{\bar{\alpha}-\alpha} \leq 2\frac{1-(\sqrt{2})^{\bar{\alpha}-1}}{\sqrt{2}}\bar{\beta}q_{t*}^{\bar{\alpha}-\alpha}, \tag{104}$$

where the first inequality follows from (102), (103), and $\beta_t \geq \beta_1 \geq 1$, the second inequality from the definition of $\bar{\beta}$, the third inequality from $\min\{1 - \frac{3}{2}\alpha, 1 - 2\alpha\} \geq \bar{\alpha} - \alpha$ since $\bar{\alpha} = 1 - \alpha$, and the last inequality from $1 - x \leq (1 - (\sqrt{2})^{x-1})/\sqrt{2}$ for $x \leq 1$. Thus the condition (59) is satisfied. Therefore, from Lemma 15, we have $h_{t+1} = H_\alpha(q_{t+1}) \leq 2H_\alpha(q_t) = 2h_t$, which implies that Assumption (iii) in (12) is satisfied.

Finally, we check the assumption (14) in Theorem 7. We first consider the first inequality in (14). From the definition of $z_t$ and the fact that $q_{ti} \leq q_{t\tilde{I}_t}$ for $i \neq \tilde{I}_t$, we get

$$
\begin{aligned}
z_t &= \frac{4ck}{1-\alpha}\left\{\sum_{i \neq \tilde{I}_t} q_{ti}^{2-\alpha} + \left(\min\{q_{t\tilde{I}_t}, 1 - q_{t\tilde{I}_t}\}\right)^{2-\alpha}\right\} \\
&\leq \frac{4ck}{1-\alpha}\left\{\sum_{i \neq \tilde{I}_t} q_{ti}^{2-\alpha} + \left(\sum_{i \neq \tilde{I}_t} q_{ti}\right)^{2-\alpha}\right\} \\
&\leq \frac{8ck}{1-\alpha}\left(\sum_{i \neq \tilde{I}_t} q_{ti}\right)^{2-\alpha} \leq \frac{8ck}{1-\alpha}\left(\sum_{i \neq a^*} q_{ti}\right)^{2-\alpha} = \frac{8ck}{1-\alpha}(1 - q_{ta^*})^{2-\alpha},
\end{aligned} \tag{105}
$$

where the second inequality holds from the inequality $x^a + y^a \leq (x+y)^a$ for $x, y \geq 0$ and $a \geq 1$, and the third inequality from $q_{ti} \leq q_{t\tilde{I}_t}$. Hence, combining (105) and the upper bound on $h_t$ in (70), we obtain

$$
z_t h_t \leq \frac{8ck}{1-\alpha}(1 - q_{ta^*})^{2-\alpha} \cdot \frac{1}{\alpha}(k-1)^{1-\alpha}(1 - q_{ta^*})^\alpha = \underbrace{\frac{8ck(k-1)^{1-\alpha}}{\alpha(1-\alpha)}}_{=\rho_1}(1 - q_{ta^*})^2. \tag{106}
$$

We next consider the second inequality in (14). We can upper bound $u_t$ as

$$
\begin{aligned}
u_t &= \frac{8\max\{c, 1\}}{1-\alpha}\left(\min\{q_{t\tilde{I}_t}, 1 - q_{t\tilde{I}_t}\}\right)^{1-\alpha} \leq \frac{8\max\{c, 1\}}{1-\alpha}\left(\sum_{i \neq \tilde{I}_t} q_{ti}\right)^{1-\alpha} \\
&\leq \frac{8\max\{c, 1\}}{1-\alpha}\left(\sum_{i \neq a^*} q_{ti}\right)^{1-\alpha} = \frac{8\max\{c, 1\}}{1-\alpha}(1 - q_{ta^*})^{1-\alpha},
\end{aligned} \tag{107}
$$

where the second inequality follows from $q_{t\tilde{I}_t} \geq q_{ti}$ for all $i \neq \tilde{I}_t$. Hence, combining the last inequality with (70),

$$
u_t h_t \leq \underbrace{\frac{4\max\{c, 1\}(k-1)^{1-\alpha}}{\alpha(1-\alpha)}}_{=\rho_2}(1 - q_{ta^*}). \tag{108}
$$

Hence, the assumption (14) is satisfied with above $\rho_1$ and $\rho_2$, and thus we have completed the proof. $\qquad \square$

