# OpenReview forum: "A Simple and Adaptive Learning Rate for FTRL in Online Learning with Minimax Regret of $\Theta(T^{2/3})$ and its Application to Best-of-Both-Worlds"
_NeurIPS.cc/2024/Conference — NeurIPS 2024 poster_

### Official Review · Reviewer_gjM7 · 2024-06-17

**Soundness:** 3
**Presentation:** 2
**Contribution:** 2
**Rating:** 6
**Confidence:** 2

**Summary:**

This paper provides a new adaptative learning rate framework for hard problems, called Stability-Penalty-Bias matching learning rate (SPB-matching). Using this SPB-matching learning rate, the paper proposes a Best-of-both-worlds (BOBW) algorithm framework for hard online learning problems. It not only achieves simultaneous optimality in the stochastic and adversarial regimes but also achieves the MS-type bound in the adversarial regime with a self-bounding constraint. The paper further shows the utility of their frameworks by studying two hard problems: partial monitoring and graph bandits.

**Strengths:**

- **Clear structure**: The paper is well-organized, starting with the establishment of the framework for the learning rate, followed by the introduction of an algorithmic framework based on this learning rate, and finally exploring two applications of their frameworks.
- **Relatively comprehensive study**: The paper considers both the stochastic and adversarial regimes, as well as an intermediate regime termed the adversarial regime with a self-bounding constraint. It provides algorithms and theoretical guarantees, along with specific parameter choices in Algorithm 1 for two particular problems.

**Weaknesses:**

- **Need more background introduction**: The authors should consider including an optimization problem aligned with (1) to better illustrate the hard problems. Additionally, more explanation is needed for the terms $z_t, h_t, u_t$ in the introduction or preliminaries sections to enhance understanding.
- **Strong assumptions**: Although the assumptions in Theorem 7 are checked for two specific problems, they are quite strong and may not hold for many hard problems. The authors should add more discussion about these assumptions, explaining the meaning of each assumption and identifying scenarios in which these assumptions hold.

**Questions:**

- Can the authors provide more explanations on the terms $z_t$, $h_t$, and $u_t$?
- On page 5, line 146, the paper assumes access to $\hat{h}_t$, which upper bounds $h_t$. Why is this always the case in reality?
- Why does Theorem 7 not depend on $p_0$?
- Why are $z_t$ and $u_t$ chosen as they are in the two examples? What is the practical meaning behind this choice? How should $z_t$ and $u_t$ be decided for other examples?
- Typo: In Lemma 5, line 163, it should be $J \in \mathbb{N}$ instead of $j \in \mathbb{N}$.

**Limitations:**

See weaknesses.

---

> ### Author Rebuttal · Authors · 2024-08-07
>
> We are grateful for your valuable time and detailed review.
> Below are our responses to the review.
>
> > The authors should consider including an optimization problem aligned with (1) to better illustrate the hard problems.
>
> Thank you for the comment regarding optimization problems.
> But, we may not understand the reviewer's intent, so if you have time, we would appreciate it if you could be more specific about what optimization problems should be included.
>
> > Additionally, more explanation is needed for the terms $z_t, h_t, u_t$ in the introduction or preliminaries sections to enhance understanding.
>
> Thank you for your suggestion.
> We will add an intuitive explanation of $z_t$ and $h_t$ in the introduction, along with an example in the expert setting.
> As for $u_t$, it is a parameter that depends on the problem setting, and we provided examples in the sections on partial monitoring and graph bandits in the main text.
>
>
> > Although the assumptions in Theorem 7 are checked for two specific problems, they are quite strong and may not hold for many hard problems. The authors should add more discussion about these assumptions, explaining the meaning of each assumption and identifying scenarios in which these assumptions hold.
>
> Thank you for your comment. From a high-level perspective, these assumptions hold even for partial monitoring, which includes many sequential decision-making problems as special cases, as confirmed in Section 5. In this sense, we do not consider them to be strong assumptions.
>
> Below, we provide a detailed explanation of each assumption:
> Assumptions (i) and (ii) naturally arise in the analysis of FTRL. In particular,
> Assumption (i): This is very standard because this is satisfied if we use an unbiased estimator $\hat{\ell}$. For example, it can be found in Chapter 11 of Lattimore and Szepesvari's book [36].
> Assumption (ii): This is an upper bound on the stability term of FTRL, which can be controlled if the magnitude of the loss estimator is bounded favorably, and it is standard assumption in the analysis of bandit algorithms. For example, it can be found in Chapter 29 of Lattimore and Szepesvari's book [36].
> Assumption (iii): This can be satisfied by ensuring the stability of the output of FTRL and has been investigated in various settings, for example, [55,26,28] and (Bubeck et al. 2018).
> For these reasons, Assumptions (i), (ii), and (iii) are considered relatively mild.
> The second condition in Eq. (14) has been used in existing research on best-of-both-worlds [26,28]. The first condition in Eq. (14) is new but can be seen as a natural extension of the second condition in Eq. (14) to problems with the minimax regret of $\Theta(T^{2/3})$.
>
> - Bubeck et al. Sparsity, variance and curvature in multiarmed bandits. ALT 2018.
>
> > On page 5, line 146, the paper assumes access to $\hat{h}_t$, which upper bounds $h_t$. Why is this always the case in reality?
>
> Thank you for your valuable comment to improve the paper.
> From Assumption (iii) we have $h_t \leq c h_{t-1}$ for some constant $c$.
> Hence if we set $\hat{h}\_t \leftarrow c h_{t-1}$, we have $h_t \leq \hat{h}\_t$.
> Note that $h\_{t-1}$ can be calculated from the information available at the end of round $t-1$, so it can be used when determining $\beta_t$.
> In the revised version, we will include this discussion.
>
>
> > Why does Theorem 7 not depend on $p_0$?
>
> The parameter $p_0$ is a parameter that should be appropriately determined according to the problem setting.
> We design $p_0$ in Algorithm 1 to satisfy Assumption (ii). For example, in partial monitoring, we set $p_0 = 1/k$, and in graph bandits, we set $p_0 = u$ (see Line 265).
> To improve the current text, we will revise the statement in Theorem 7 to, "If Algorithm 1 satisfies Assumptions (i)--(iii), then its regret is bounded as ...".
>
> > Why are $z_t$ and $u_t$ chosen as they are in the two examples? What is the practical meaning behind this choice? How should $z_t$ and $u_t$ be decided for other examples?
>
> The values of $z_t$ and $u_t$ are determined by the problem setting and the regularizer.
> In general, $u_t$ is chosen to satisfy the condition for stability (the assumption of Lemma 14 in our paper). Controlling the magnitude of the loss estimator by adding appropriate exploration to manage the stability term is common in the literature of FTRL in bandit problems (e.g., [8,26,54] and [36, Chapter 27]).
>
> Regarding $z_t$, it is difficult to provide a general determination strategy within the limited space, as it varies depending on the form of the adaptive bounds that the algorithm designer aims to achieve. However, in the context of best-of-both-worlds bounds, it is common for $z_t$ to be determined based on the output of FTRL, $q_t$, to satisfy conditions similar to Eq. (14) in Theorem 2 [61,62,55,26,28].

---

> > ### Comment · Reviewer_gjM7 · 2024-08-07
> >
> > Thank you for your response to my comments.
> > The explanation provided by the authors addresses my concerns about the strong assumptions, so I will raise my score to 6. I hope the authors will include an intuitive explanation of $z_t$ and $h_t$ in the revised version.

---

### Official Review · Reviewer_7rt1 · 2024-06-30

**Soundness:** 3
**Presentation:** 2
**Contribution:** 3
**Rating:** 6
**Confidence:** 3

**Summary:**

The paper aims to develop a new adaptive learning rate framework for the Follow-the-Regularized-Leader (FTRL) algorithm that addresses online learning problems with a minimax regret of $\Theta(T^{2/3})$.

It specifically targets problems with indirect feedback, such as partial monitoring and graph bandits, and demonstrates the efficacy of the proposed framework in improving Best-of-Both-Worlds (BOBW) regret upper bounds.

**Strengths:**

1. **Innovative Approach:** The SPB-matching learning rate is a novel contribution that simplifies the design of adaptive learning rates for complex online learning problems. It has the potential to be applied to other settings.
2. **Unified Framework:** It successfully unifies the BOBW guarantee for hard problems, representing a good theoretical advancement. Its application to partial monitoring and graph bandits provides tangible improvements over existing methods.

**Weaknesses:**

1. **Practical Consideration:** The effectiveness of the SPB-matching framework depends on the proper tuning of parameters, which might limit its practical applicability without further optimization techniques. Furthermore, there are no experiments, as a result of which, it is unclear how the proposed method works in practice, even in the simulation.
2. **Clarity and Accessibility:**  The paper is difficult to follow due to its technical nature. Some equations and results are presented without sufficient explanation, and the extension to the two case studies introduces many new concepts that are hard to follow.

**Questions:**

1. **Extension to High-dimensional Sparse Bandits:** The $\Theta(T^{2/3})$ regret reminds me of the results in high-dimensional sparse linear bandits [1*]. The regret in [1*] is $\Theta(s^{1/3}T^{2/3})$ (where $s$ is the level of sparsity), and the algorithm in [1*] also depends on a sort of "forced exploration." Is it possible to apply the framework developed in this paper to this high-dimensional and sparse setting?

    -  [1*] Hao, Botao, Tor Lattimore, and Mengdi Wang. "High-dimensional sparse linear bandits." Advances in Neural Information Processing Systems 33 (2020): 10753-10763.

2. **Other Upper Bounds:** The introduction of the minimax regret $\Theta(T^{2/3})$ is somewhat abrupt. It makes me wonder whether in some cases, there is a minimax regret like $\Theta(T^{a/(a+1)})$ for some parameter $a>0$ and why the case $a=2$ is so special that we should focus on it.

3. **General Lower Bound:** Is there any general lower bound for the regret in Theorem 2? While particular cases may have existing lower bounds, a general tight lower bound is expected given that the paper starts from a general framework and provides a general upper bound.


4. **Clarification on Exploration Rate Requirement:** Could the author explain why $\gamma_t \ge u_t/\beta_t$ (in line 138) is needed? In which lemma or proof is this condition necessary?

5. **Comparison of SPB-Matching Rules:** Which rule is better in theory or practice: Rule 1 or Rule 2 in (6)?

6. **High-Level Explanation of Regret Improvement:** Could the author briefly explain why the regret is improved after applying the established framework in Sections 5 and 6? Is there any high-level explanation beyond the technical details?

**Limitations:**

See the Weaknesses and Questions.

---

> ### Author Rebuttal · Authors · 2024-08-07
>
> Thank you very much for taking the time to carefully review our paper and for providing many questions. Below are our responses to your review.
>
> > The effectiveness of the SPB-matching framework depends on the proper tuning of parameters, which might limit its practical applicability without further optimization techniques.
>
> Thank you for your comment about practical applicability.
> However, we may not fully understand the reviewer's intent, so if you have time, we would appreciate it if you could be more specific about what further optimization techniques mean.
>
> If the further optimization problem refers to the optimization of the fractional domination number in graph bandits as described in Eq. (19), note that this optimization problem can be computed efficiently by solving a linear programming problem once before the game begins [13].
>
> Additionally, note that the tuning of parameters can be determined based on past observations, without the need to solve any optimization problem other than FTRL. This is standard in many best-of-both-worlds algorithms (e.g., [26,54,62]).
>
>
> > there are no experiments, as a result of which, it is unclear how the proposed method works in practice, even in the simulation.
>
> Thank you for your comment. This paper focuses on theoretical aspects, and investigating the numerical performance in simulations or practical scenarios is an important future work.
>
> > ... there is a minimax regret like $\Theta(T^{a/(a+1)})$ for some parameter $a > 0$ and why the case $a=2$ is so special ...
>
> Thank you for pointing out this important point.
> The case where $a=2$ is special due to the classification theorem in partial monitoring. Partial monitoring is a very general problem that includes a wide range of sequential decision-making problems as special cases. It is known that, depending on the relationship between the loss matrix $\mathcal{L}$ and the feedback matrix $\Phi$, the minimax regret can be classified into one of four categories: $0$, $\Theta(\sqrt{T})$, $\Theta(T^{2/3})$, or $\Omega(T)$ (see line 484). Among these, the classes with non-trivial difficulties and particular importance are the problems with a minimax regret of $\Theta(\sqrt{T})$ and $\Theta(T^{2/3})$.
> This paper focuses on the problems with a minimax regret of $\Theta(T^{2/3})$ due to this classification theorem.
> In the revised version, we will explain why we particularly focus on the $\Theta(T^{2/3})$ case.
>
> > Is there any general lower bound for the regret in Theorem 2? While particular cases may have existing lower bounds, a general tight lower bound is expected given that the paper starts from a general framework and provides a general upper bound.
>
> Under the general online learning setup given in Section 2, it is almost impossible to prove a lower bound for Theorem 2.
> This general online learning setup is extremely general.
> For instance, it includes partial monitoring as a special case. However, partial monitoring itself is very complex, and even when focusing solely on the adversarial regret, no lower bound that depends on variables other than $T$ is known (see Item 14 of Section 37.9 in Lattimore and Szepesvari's book [36]).
> Since constructing a lower bound within the general online learning framework is far more challenging than considering a lower bound for partial monitoring, it is almost impossible to prove a lower bound for Theorem 2.
>
> Still, the results for graph bandits (Corollary 9) obtained as a special case of Theorem 2 match the lower bound. Additionally, as mentioned in Global Comments, in the setting of multi-armed bandits with paid observations, the upper bound obtained by SPB-matching in the adversarial regime also matches the lower bound. From these observations, we can see that Theorem 2 matches the lower bound in several problem settings.
>
> > Could the author explain why $\gamma_t \geq u_t / \beta_t$ (in line 138) is needed? In which lemma or proof is this condition necessary?
>
> This condition is used to bound the magnitude of the loss estimator $\hat{\ell}_t$.
> For example, in partial monitoring it is used in Eq. (61), and in graph bandits it is used in Eq. (79). Thanks to this lower bound of $\gamma_t$, we can apply Lemma 14 to control the stability term of FTRL (The LHS of (56)).
> In the revised version, we will provide a more detailed explanation in line 138 about why this condition is necessary.
>
> Such efforts to bound the magnitude of the loss estimator are not unique to this paper but frequently appear in FTRL-related analyses. For instance, in the Exp3 algorithm [8], which is equivalent to FTRL with Shannon entropy, they employ exploration to bound the magnitude of the *gain* vector.
>
> > Could the author briefly explain why the regret is improved after applying the established framework in Sections 5 and 6? Is there any high-level explanation ...?
>
> The high-level reason for the improvement is that the introduction of SPB-matching allows us to use the Tsallis entropy regularizer with an appropriate exponent $\alpha$. Such improvements have been known in various contexts, such as multi-armed bandits [3,5], sparse bandits (Kwon and Perchet, 2016), and (strongly observable) graph bandits (Zimmert and Lattimore, 2019 and [26]).
> To achieve best-of-both-worlds using Tsallis entropy, we need to derive a regret upper bound that depends simultaneously on stability and penalty components [26,28,55].
> Our paper makes this possible even for problems with a minimax regret of $\Theta(T^{2/3})$.
>
> - Kwon and Perchet. Gains and losses are fundamentally different in regret minimization: The sparse case, JMLR 2016.
> - Zimmert and Lattimore. Connections between mirror descent, Thompson sampling and the information ratio, NeurIPS 2019.
>
> Due to space constraints, we will respond to the items that are considered to have a relatively small impact on the evaluation in the following Comments.

---

> ### Author Response · Authors · 2024-08-07
> **Additional Replies**
>
> Here, we will respond to the items that are considered to have a relatively small impact on the evaluation, which cannot be included in the above rebuttal due to the space constraint.
>
> > Is it possible to apply the SPB-matching framework developed in this paper to high-dimensional sparse linear bandits (by Hao-Lattimore-Wang 2020)?
>
> Thank you for your suggestion.
> The paper by Hao, Lattimore, and Wang (2020) on high-dimensional sparse linear bandits considers the stochastic regime.
> To the best of our knowledge, there are no studies investigating whether a dimension-free bound is possible in the adversarial regime similar to Hao, Lattimore, and Wang (2020).
> Therefore, before investigating the applicability of our SPB-matching framework in this setting, it is necessary to study high-dimensional sparse linear bandits in the adversarial regime.
>
> > Which rule is better in theory or practice: Rule 1 or Rule 2 in (6)?
>
> First, the information available for determining $\beta_t$ differs between Rule 1 and Rule 2. In Rule 1, it is assumed that the information up to time $t-1$, $z_t$, $u_t$, and $\hat{h}_t \geq h_t$ are known when determining $\beta_t$. In Rule 2, it is assumed that the information up to time $t-1$ and $\hat{h}_t \geq h_t$ are known when determining $\beta_t$. Rule 1 is included due to theoretical interest and is not used in Sections 4 and 5. In the revised version, we will explicitly state this and further emphasize that the information available when determining $\beta_t$ differs between Rule 1 and Rule 2.

---

### Official Review · Reviewer_TQWF · 2024-07-07

**Soundness:** 2
**Presentation:** 2
**Contribution:** 2
**Rating:** 4
**Confidence:** 2

**Summary:**

In this work, the authors propose a simple and adaptive learning rate for FTRL, which can achieve the minimax regret of $O(T^{2/3})$ for some hard online learning problems. Specifically, they have applied their algorithm (FTRL with the proposed learning rate) to achieve the best-of-worlds regret bounds for partial monitoring (with global observability) and graph bandits (with weak observability), respectively. For the first problem, compared with the best existing results, this paper achieves an $\log(T)/\log(k)$ improvement in the stochastic setting, and an $(\log(T)+\log^2(T)/\log(k))^{1/3}$ improvement in the adversarial setting. But, for the second problem, compared with the best existing results, it seems that this paper can only achieve worse regret bounds (by a factor of $\delta^\ast/\delta$) in the stochastic and adversarial settings.

**Strengths:**

Compared with previous studies, this paper has the following strengths.
1) A simple and adaptive learning rate is proposed for FTRL with forced exploration, which can achieve the minimax regret for some hard online learning problems, such as partial monitoring (with global observability) and graph bandits (with weak observability).
2) The authors have improved the best-of-worlds regret bounds for partial monitoring by utilizing the proposed learning rate. Specifically,  compared with the best existing results, this paper achieves an $\log(T)/\log(k)$ improvement in the stochastic setting, and an $(\log(T)+\log^2(T)/\log(k))^{1/3}$ improvement in the adversarial setting.

**Weaknesses:**

However, I have some concerns about this paper.
1) The proposed learning rate is highly inspired by an existing adaptive learning rate in a recent work [26]. Although the existing one can only be utilized to achieve the minimax regret bound of $O(\sqrt{T})$ for some relatively easy problems, the only difference for the proposed learning rate is to further trade off a bias term (with the stability term), which does not bring enough challenges.
2) The motivation for such a learning rate for FTRL is to deal with some hard online learning problems (e.g., partial monitoring and graph bandits) that have been well-studied before. Although this paper has applied their algorithm (FTRL with the proposed learning rate) to achieve the best-of-worlds regret bounds for partial monitoring and graph bandits respectively, it does not bring significant advantages compared with existing results, i.e., only logarithmic improvements for partial monitoring and even worse results for graph bandits.
3) In Section 3, two update rules are proposed for the simple and adaptive learning rate. However, in Sections 4 and 5, it seems that only the second update rule is utilized. So, it is confusing why the authors propose the first update rule and spend non-ignorable spaces in the main text to introduce the corresponding results.
4) The writing of this paper needs to be improved. For example, the majority of the literature review is only proposed in the appendix, which indeed reduces the readability of this paper. In addition, in line 273, the authors emphasize that "Our bound is the first BOBW FTRL-based algorithm with the $O(\log T)$ bound ...". Although this is not wrong, it is not much to be particular about, because there already exist other non-FTRL-based algorithms to achieve the  $O(\log T)$ regret bound.

**Questions:**

Besides the concerns discussed above, I also have the following two questions.
1) Can the authors provide a detailed comparison between the proposed algorithm and existing algorithms in works [26, 54]?
2) It seems that Dann et al. [15] propose a black-box approach to best-of-worlds regret bounds in bandits. Can their approach be utilized to solve the partial monitoring problem?

**Limitations:**

The authors have provided some discussions on the limitations of this work.

---

> ### Author Rebuttal · Authors · 2024-08-07
>
> We greatly appreciate your thorough and thoughtful review of our work. Here are our responses to your feedback.
>
> > The proposed learning rate is highly inspired by an existing adaptive learning rate in a recent work [26]. ...  which does not bring enough challenges.
>
> Yes, as the reviwer pointed out, the proposed adaptive learning rate is highly inspired by [26].
> The primary challenge in our paper lies in finding a beneficial form of regret upper bounds that simultaneously depend on both stability and penalty for problems with a minimax regret of $\Theta(T^{2/3})$.
> We found that the form of $O\Big( \big(\sum_{t=1}^T \sqrt{z_t h_{t+1} \log T} \big)^{1/3} \Big)$ is beneficial. To our knowledge, this form of regret upper bound is new.
> In the revised version, we will include a discussion on these points.
> Due to the space constraint, please refer to *primary challenge in our paper* in Global Comments for further details.
>
> > ... only logarithmic improvements for partial monitoring and even worse results for graph bandits.
>
> and
>
> > ... the authors emphasize that "Our bound is the first BOBW FTRL-based algorithm with the $O(\log T)$ bound ...". it is not much to be particular about, ...
>
>
> **Logarithmic improvement**
>
> First, logarithmic improvement is important. Historically, significant efforts have been dedicated to achieving an $O(\sqrt{kT})$ regret upper bound without the logarithmic term in multi-armed bandits [3,5]. Attempts to improve this logarithmic term have subsequently led to the development of the notable best-of-both-worlds algorithm in multi-armed bandits [61]. Furthermore, beyond multi-armed bandits, there are numerous efforts aimed at improving the logarithmic factor to enhance performance, for example, (Kwon and Perchet 2016; Zimmert and Lattimore 2019; Eldowa et al. 2023). Based on these points, we can see that the improvement of the logarithmic factor is important.
>
> References
> - Kwon and Perchet. Gains and losses are fundamentally different in regret minimization: The sparse case, JMLR 2016.
> - Zimmert and Lattimore. Connections between mirror descent, Thompson sampling and the information ratio, NeurIPS 2019.
> - Eldowa et al. On the Minimax Regret for Online Learning with Feedback Graphs, NeurIPS 2023.
>
>
>
> **Comparison with graph bandits result in Dann et al. [15]**
>
> As pointed out, the regret upper bound for our graph bandits indeed deteriorates compared to the bound of Dann et al. [15] (this difference arises because the (fractional) weak domination number in Dann et al. becomes the fractional domination number in our bound, and note that this change only worsens the bound when the feedback graph has a self-loop).
>
> However, the approach of Dann et al. [15] employs a highly complex, multi-stage reduction approach. Moreover, it has the disadvantage of discarding past observations, similar to the doubling-trick.
> We have demonstrated that the framework of Follow-The-Regularized-Leader alone can achieve an upper bound similar to that accomplished by this highly complex approach, which is a significant theoretical advancement.
>
> **Generality to other problem settings**
> Using our SPB-matching, we believe that best-of-both-worlds algorithms can be developed for a wide range of problems with a minimax regret of $\Theta(T^{2/3})$.
> For instance, the authors found after the submission that it is possible to achieve best-of-both-worlds bounds in multi-armed bandits with paid observations [53] by SPB-matching.
> Due to the space constraint, please refer to *Generality to Other Problem Settings* in Global Comments for further details.
>
>
> > ... two update rules are proposed for the simple and adaptive learning rate. However, in Sections 4 and 5, it seems that only the second update rule is utilized. So, it is confusing ...
>
> Thank you for your suggestion.
> Rule 1 was included for theoretical interest, and as the reviewer pointed out, it is not used in Sections 4 and 5.
> In the revised version, we will emphasize that only Rule 2 is used in Sections 4 and 5.
>
> > ... the majority of the literature review is only proposed in the appendix, ...
>
> Due to page constraints, much of the literature review had to be included in the appendix. In the revised version, we will aim to include as much of the literature review as possible in the main text.
>
> Below, we provide replies to the questions.
>
> > It seems that Dann et al. [15] propose a black-box approach to best-of-worlds regret bounds in bandits. Can their approach be utilized to solve the partial monitoring problem?
>
> Thank you for your question. By employing their black-box reduction approach, it seems possible to achieve an upper bound of the same order as our upper bound in globally observable partial monitoring.
> Apologies for not realizing this at the time of submission. This is because Dann et al.'s method cannot be used in partial monitoring games with minimax regret of $\sqrt{T}$ unless the loss of the selected action is observed [56].
> Nevertheless, as previously mentioned, the approach by Dann et al. [15] is a complicated approach involving multi-stage reductions and has the drawback of discarding past observations, similar to the doubling-trick.
> Hence, demonstrating that using the FTRL framework alone can achieve the same upper bound is a significant theoretical advancement.
> In the revised version, we will include the regret upper bound obtained by Dann et al. in Table 1 and describe the differences between their method and the advantages of using FTRL directly.
>
> Due to space constraints, we will respond to the items that are considered to have a relatively small impact on the evaluation in the following Comments.

---

> > ### Comment · Reviewer_TQWF · 2024-08-09
> >
> > Thanks for the authors' response. I currently do not have additional questions, and will make my final decision after further discussing with other reviewers and AC.

---

> ### Author Response · Authors · 2024-08-07
> **Additional Replies**
>
> Here, we will respond to the items that are considered to have a relatively small impact on the evaluation, which cannot be included in the above rebuttal due to space constraints.
>
>
> > What are the lower bounds for partial monitoring with global observability?
>
> To the best of our knowledge, beyond the dependency on $T$, little is known about the lower bound for partial monitoring (with global observability).
> In the adversarial regime, the dependency on $T$ is known to be $\Omega(T^{2/3})$ [9, 34, 35].
> However, little is understood about the dependencies on the variables $k$, $d$, and $m$, which is also mentioned in Section 37.9 of [36].
> In the stochastic regime, an asymptotic distribution-dependent lower bound of $\Omega(D (\nu^* )\log T)$ is known [30], where $D(\nu^*)$ denotes the optimal value of a certain complex optimization problem.
> However, it is not clear how this $D(\nu^*)$ depends on $k$, $d$, and $m$, and investigating this is important future work.
> In the revised version, we will discuss that little is known about the dependencies other than $T$ for the lower bound in partial monitoring.
>
> > Can the authors provide a detailed comparison between the proposed algorithm and existing algorithms in works [26, 54]?
>
> (Comparison with [26])
> As mentioned above, this study considers problems with the minimax regret of $\Theta(T^{2/3})$, and an ideal form dependent on the stability component $z_t$ and penalty component $h_t$ are different.
>
> (Comparison with [54])
>
> In [54]:
> - Their main regularizer is the negative Shannon entropy.
> - The regret upper bound in the stochastic regime is $O((\log T)^2)$.
> - Their learning rate (for globally observable games) is highly complicated.
>
> Our approach:
> - Our main regularizer is the negative Tsallis entropy.
> - The regret upper bound in the stochastic regime is $O(\log T)$.
> - The learning rate is designed based on a simple principle that matches stability, penalty, and bias components.

---

### Official Review · Reviewer_YZuC · 2024-07-09

**Soundness:** 3
**Presentation:** 3
**Contribution:** 3
**Rating:** 6
**Confidence:** 3

**Summary:**

This paper considers the online problems with a minimax regret of $\Theta(T^{2/3})$ and proposes a new learning rate tuning based on the FTRL algorithm to tackle these problems. The development of the learning rate is straightforward and simpler compared to previous works. The proposed algorithm is versatile in various applications, and the authors prove that it can achieve improved best-of-both-worlds regret bounds.

**Strengths:**

- This paper is technical solid though the discussion for the techniques is easy-to-follow.
- It proposes a simpler and straightforward tuning strategy, which leads to better best-of-both-world theoretical guarantees in partial monitoring and graph bandit problems.
- The applications justify their findings.

**Weaknesses:**

The main technique of designing learning rates so that the stability and bias terms are matched seems to already appear in the cited paper [26], and the development of the method seems to focus on how to solve the additional term introduced by the 'forced exploration' strategy. With a quick glance at the cited paper [26], in Section 5.1, this paper also employs the 'forced exploration' strategy, except that the balance ratio $\gamma$ is fixed. It would be beneficial to further discuss the difficulties encountered when tackling the 'hard problem'.

**Questions:**

Could the author respond to the concerns in the 'Weaknesses'?

---

> ### Author Rebuttal · Authors · 2024-08-06
>
> We appreciate your valuable time and helpful comments.
> Below is our response to your comments.
>
> > With a quick glance at the cited paper [26], in Section 5.1, this paper also employs the 'forced exploration' strategy, except that the balance ratio is fixed. It would be beneficial to further discuss the difficulties encountered when tackling the 'hard problem'.
>
> The primary difficulty in our paper lies in finding a beneficial form of regret upper bounds that simultaneously depend on both stability and penalty for problems with a minimax regret of $\Theta(T^{2/3})$.
> For problems with a minimax regret of $\Theta(\sqrt{T})$, it is known that the form of $O\Big(\sqrt{\sum_{t=1}^T z_t h_{t+1} \log T}\Big)$ for the stability component $z_t$ and the penalty component $h_t$ allows us to derive best-of-both-worlds bounds [26, 28, 55].
> However, for problems with a minimax regret of $\Theta(T^{2/3})$, it is non-trivial to determine the form of regret upper bounds that are useful for achieving best-of-both-worlds bounds.
> We found that the form of $O\Big( \big(\sum_{t=1}^T \sqrt{z_t h_{t+1} \log T} \big)^{1/3} \Big)$ is beneficial. To our knowledge, this form of regret upper bound is new.
> In the revised version, we will include a discussion on these points.
>
> Regarding the difference on the exploration rate from Section 4.1 of [26] (we believe Section 5.1 is a typo for Section 4.1), they only consider problems with a minimax regret of $\Theta(\sqrt{T})$, and they set the exploration rate $\gamma_t$ either to $\gamma_t = 0$ or $\gamma_t \simeq z_t / \beta_t$.
> When $\gamma_t \simeq z_t / \beta_t$, the additional regret due to this exploration is of the same order as the stability term, and thus a tuning strategy of the learning rate $\beta_t$ remains the same.
> In contrast, in problems with a minimax regret of $\Theta(T^{2/3})$, due to the nature of handling indirect feedback, forced exploration becomes necessary. As a result, the magnitude of the loss estimator increases, resulting in $\gamma_t$ appearing in the denominator of the stability term (see Lines 134--140). Hence, if we use $\gamma_t \simeq z_t / \beta_t$, the stability term becomes excessively large. Setting a larger exploration rate, $\gamma_t \simeq \sqrt{z_t / \beta_t}$, resolves this issue.

---

> > ### Comment · Reviewer_YZuC · 2024-08-12
> >
> > Thank you for your responses. I will keep my positive score.

---

### Official Review · Reviewer_5JDu · 2024-07-15

**Soundness:** 4
**Presentation:** 3
**Contribution:** 3
**Rating:** 7
**Confidence:** 4

**Summary:**

The paper introduces a novel adaptive learning rate framework for Follow-the-Regularized-Leader (FTRL) tailored to online learning problems characterized by a minimax regret of Θ(T^2/3). This new learning rate, termed Stability-Penalty-Bias (SPB) matching, is designed by balancing stability, penalty, and bias terms within the decomposition of the regret as $\text{Reg}\_T \\leq \\sum\_{t=1}^T \\frac{z\_t}{\\beta\_t \\gamma\_t} + \\beta\_1 h\_1 + \\sum\_{t=2}^T (\\beta\_t - \\beta\_{t-1}) h\_t + \\sum\_{t=1}^T \\gamma\_t$. The paper demonstrates the efficacy of this approach through its application to two significant “hard” online learning problems: partial monitoring and graph bandits, both of which involve indirect feedback. The proposed framework improves upon existing Best-of-Both-Worlds (BOBW) regret bounds, providing simpler yet effective learning rates compared to existing methods and consequently achieving $O(\\log T)$ regret in the stochastic regime and $O(T^{2/3})$ in the adversarial regime.

**Strengths:**

This work proposes a novel and interesting unified framework for analyzing regret in online learning problems, inspired by previous work with similar methods of regret decomposition.
This enables new BOBW guarantees for the problems of partial monitoring with global observability and graph-feedback bandits with weak observability, significantly improving over previous work.
The main results are also introduced with sufficient generality that can help the application and adoption of the proposed SPB-matching technique to other “hard” problems as suggested by the authors.
Additionally, the same method is shown to retrieve regret bounds under the more general setting with self-bounding constraints.

The algorithm is simple to understand for readers who have familiarity with the related work. It is a variant of the well-known follow-the-regularized-leader (FTRL) with additional exploration (given the weak observability of the considered problems), negative Tsallis entropy regularizer (with an appropriate tuning of its parameter $\\alpha = 1-1/(\\log k)$), and careful tuning of the learning rate via the proposed SPB-matching.
Even considering the simplicity of the main methodology, the observations and ideas behind it are nontrivial and clever.
They required some care to have all the pieces fit together to achieve the final results.

A secondary but nontrivial observation made in this work is that the generality of the Tsallis entropy provides an improvement to the regret bounds compared to using only Shannon entropy and log-barrier.
A similar phenomenon has already been noted in previous work, but to the best of my knowledge, this is the first time this observation has been made for the problems considered here.

Finally, the presentation slowly introduces the SPB-matching technique and the ideas behind it help understand how the algorithm works for providing the desired guarantees.
This might also help in adopting similar techniques in other settings, possibly providing new and improved BOBW results in additional “hard” online learning problems.

**Weaknesses:**

There is a clear effort by the authors to provide a smooth and clear description of the SPB-matching technique.
However, the large number of parameters made it hard to fully understand how each would exactly influence the analysis of the final regret bound of the algorithm.
Only after multiple reads did the full details click and it was clear how all the pieces nicely fit together.
In any case, this appears to be caused by the generality of this unified framework and is likely to be unavoidable.

Furthermore, the regret bounds in Corollaries 9 and 11 are not fully clear because of the presence of $\\kappa$ and $\\kappa’$.
Given that all the other parameters have specific values for the two problems considered here, and both $\\kappa$ and $\\kappa’$ are a function of them, replacing them and thus having an explicit dependence on the actual parameters of the problems could make it easier to understand the regret guarantees in the two corollaries and to compare them with previous related results.
Even a simple comment on the influence of $\\kappa$ and $\\kappa’$ to the regret bound for in these two specific cases would help the reader.

**Questions:**

- The possibility of adopting similar ideas to other “hard” online learning problems is suggested in the conclusions. However, while the hardness of the problems considered here is mainly related to the observability of the losses, other problems have different characteristics that make them “hard”. For example, the additional hardness of bandits with switching costs is about the loss being adaptive to the previous action of the learner, leading to an extra cost whenever the learner changes the action to play. On the other hand, SPB-matching is related to the presence of the bias term due to having extra exploration to tackle the limited observability. Do the authors believe a variant of SPB-matching could help with these additional problems? Or do they believe these problems would require different novel ideas? Just some vague intuition about this would be appreciated.
- What are the lower bounds for partial monitoring with global observability? I think being clearer about this would help the reader understand the quality of your regret bound for this problem.

Minor comments/typos:
- Line 211: “literature” instead of “litarature”
- Line 221: l.h.s. of the inline equation should be $\\sum\_{c=1}^k \\bigl( G(c,\\Phi\_{cx})\_{a} - G(c,\\Phi\_{cx})\_{b} \\bigr)$
- Line 695: duplicate “which is not desirable.”
- Line 700: “Shannon” instead of “Shanon”
- Lines 701-702: “dominating” instead of “domination”

**Limitations:**

The authors address potential limitations.

---

> ### Author Rebuttal · Authors · 2024-08-06
>
> Thank you very much for your valuable time and thorough review.
> Regarding the minor comments and typos, we will address and correct them in the revised version.
> Below is our response to the review:
>
> > the regret bounds in Corollaries 9 and 11 are not fully clear because of the presence of $\kappa$ and $\kappa'$. Given that all the other parameters have specific values for the two problems considered here, and both $\kappa$ and $\kappa'$ are a function of them, replacing them and thus having an explicit dependence on the actual parameters of the problems could make it easier to understand the regret guarantees in the two corollaries and to compare them with previous related results. Even a simple comment on the influence of $\kappa$ and $\kappa'$ to the regret bound for in these two specific cases would help the reader.
>
> Thank you for your valuable comments.
> For the partial monitoring problem, when we use $\beta_1 = 64 c_{\mathcal{G}}^2 / (1 - \alpha)$, which satisfies the first inequality in Eq. (17), we have $\kappa = O(c_{\mathcal{G}}^2 \log k + k^{3/2} (\log k)^{5/2})$ and
> $\kappa' = \kappa + O\left( ( c_{\mathcal{G}}^{2/3} (\log k)^{1/3} + \sqrt{c_{\mathcal{G}} \log k} ) ( \frac{1}{\Delta_{\min}^2} + \frac{C}{\Delta_{\min}} )  \right)$.
> For the graph bandit problem, when we use $\beta_1 = 64 \delta^* / (1 - \alpha)$, which satisfies the first inequality in Eq. (20), we have $\kappa = O(\delta^* \log k + k^{3/2} (\log k)^{5/2})$ and
> $\kappa' = \kappa + O\left( ( (\delta^* \log k)^{1/3} + \sqrt{\delta^* \log k} ) ( \frac{1}{\Delta_{\min}^2} + \frac{C}{\Delta_{\min}} )  \right)$.
> In the revised version, we will make the influence of $\kappa$ and $\kappa'$ explicit.
>
> > Do the authors believe a variant of SPB-matching could help with these additional problems? Or do they believe these problems would require different novel ideas? Just some vague intuition about this would be appreciated.
>
> Thank you for pointing out this important point.
> We believe that our approach may not be applicable to all problems with the minimax regret of $\Theta(T^{2/3})$ (hard problems), and as the reviewer suggests, different novel ideas might be required.
> Still, if a hard problem has a structure similar to hard graph bandits and partial monitoring problems that requires additional exploration, we believe that the best-of-both-worlds bounds can be achieved by SPB-matching.
>
> For instance, the authors found after the submission that it is possible to achieve best-of-both-worlds bounds in multi-armed bandits with paid observations [53] by SPB-matching.
> In this setting, the learner can observe the loss of any action by paying a cost, and the goal of the learner is to minimize the sum of the regret and total paid costs for observations.
> The cost of observations behaves similarly to the bias term in the stability–penalty–bias decomposition, and thus our SPB-matching learning rate can be applied.
> In particular, we can show that the sum of the regret and paid costs is roughly bounded by
>   $
>     O\big(
>       (c k \log k)^{1/3} T^{2/3}
>       +
>       \sqrt{T \log k}
>     \big)
>   $
>   in the adversarial regime and by
>   $
>     O\big(
>       \max\\{c,1\\} k \log k \log T / \Delta_{\min}^2
>     \big)
>   $
>   in the stochastic regime for the cost of observation $c$.
>   This is the first best-of-both-worlds bounds for multi-armed bandits with paid observations, and the bound for the adversarial regime is of the same order as [53, Theorem 3],  demonstrating the effectiveness of the SBP framework.
> The proof is almost the same as in the cases of graph bandits and partial monitoring. The revised version will include this discussion.
>
> > What are the lower bounds for partial monitoring with global observability?
>
> To the best of our knowledge, beyond the dependency on $T$, little is known about the lower bound for partial monitoring (with global observability).
> In the adversarial regime, the dependency on $T$ is known to be $\Omega(T^{2/3})$ [9, 34, 35].
> However, little is understood about the dependencies on the variables $k$, $d$, and $m$, which is also mentioned in Section 37.9 of [36].
> In the stochastic regime, an asymptotic distribution-dependent lower bound of $\Omega(D (\nu^* )\log T)$ is known [30], where $D(\nu^*)$ denotes the optimal value of a certain complex optimization problem.
> However, it is not clear how this $D(\nu^*)$ depends on $k$, $d$, and $m$, and investigating this is important future work.
> In the revised version, we will discuss that little is known about the dependencies other than $T$ for the lower bound in partial monitoring.

---

> > ### Comment · Reviewer_5JDu · 2024-08-14
> >
> > Thank you for the exhaustive responses to my questions and the useful insights. I am keeping my positive score.

---

### Author Rebuttal · Authors · 2024-08-07

Thank you very much for your valuable time and thorough, insightful reviews.
While we have replied directly to each reviewer, there are some important points that we could not fully explain due to space constraints.
Therefore, we are addressing these points as global comments here. Please feel free to let us know if you have any further questions.

**Primary challenge in our paper**

The primary challenge in our paper lies in finding a beneficial form of regret upper bounds that simultaneously depend on both stability and penalty for problems with a minimax regret of $\Theta(T^{2/3})$.
For problems with a minimax regret of $\Theta(\sqrt{T})$, it is known that the form of $O\Big(\sqrt{\sum_{t=1}^T z_t h_{t+1} \log T}\Big)$ for the stability component $z_t$ and the penalty component $h_t$ allows us to derive best-of-both-worlds bounds [26, 28, 55].
However, for problems with a minimax regret of $\Theta(T^{2/3})$, it is non-trivial to determine the form of regret upper bounds that are useful for achieving best-of-both-worlds bounds.
We found that the form of $O\Big( \big(\sum_{t=1}^T \sqrt{z_t h_{t+1} \log T} \big)^{1/3} \Big)$ is beneficial. To our knowledge, this form of regret upper bound is new.
In the revised version, we will include a discussion on these points.

**Generality to other problem settings**

Using our SPB-matching, we believe that best-of-both-worlds algorithms can be developed for a wide range of problems with a minimax regret of $\Theta(T^{2/3})$.
For instance, the authors found after the submission that it is possible to achieve best-of-both-worlds bounds in multi-armed bandits with paid observations [53] by SPB-matching.
In this setting, the learner can observe the loss of any action by paying a cost, and the goal of the learner is to minimize the sum of the regret and total paid costs for observations.
The cost of observations behaves similarly to the bias term in the stability–penalty–bias decomposition, and thus our SPB-matching learning rate can be applied.
In particular, we can show that the sum of the regret and paid costs is roughly bounded by
  $
    O\big(
      (c k \log k)^{1/3} T^{2/3}
      +
      \sqrt{T \log k}
    \big)
  $
  in the adversarial regime and by
  $
    O\big(
      \max\\{c,1\\} k \log k \log T / \Delta_{\min}^2
    \big)
  $
  in the stochastic regime for the cost of observation $c$.
  This is the first best-of-both-worlds bounds for multi-armed bandits with paid observations, and the bound for the adversarial regime is of the same order as [53, Theorem 3],  demonstrating the effectiveness of SBP-matching.
The proof is almost the same as in the cases of graph bandits and partial monitoring. The revised version will include this discussion.

---

### Decision · Program_Chairs · 2024-09-25

**Decision:**

Accept (poster)

**Comment:**

Reviewers expressed mostly positive opinions on the novelty and significance of this work. Mild doubts about the level of contribution not reaching the bar of NeurIPS were discussed but were eventually dismissed. I recommend acceptance (poster) with the caveat that the authors commit to addressing the reviewers' comments in the revised version.